# An Orthogonal Geometry-Based Algorithm for Accurate Mesoscale Eddy Detection

Yu Cai [1,2,3,4,5,†] ORCID, Jingyi Yang [1,2,3,4,†] and Jun Song [1,2,3,4,5,*]

1 Operational Oceanography Institution, Dalian Ocean University, Dalian 116023, China; caiyu@dlou.edu.cn (Y.C.); yjyxysj96@163.com (J.Y.)
2 College of Marine Science Technology and Environment, Dalian Ocean University, Dalian 116023, China
3 Liaoning Key Laboratory of Marine Real-Time Warning, Dalian 116023, China
4 Dalian Technology Innovation Center for Operational Oceanography, Dalian 116023, China
5 Dalian Xinghaiwan Laboratory, Dalian 116023, China
* Correspondence: songjun2017@dlou.edu.cn
† These authors contributed equally to this work.

## Abstract

This article introduces a mesoscale eddy detection algorithm that employs orthogonal transformations of flow field data, and subsequently, for simplicity, it is abbreviated as the OG algorithm. By implementing orthogonal geometric transformations on sea surface flow field data and examining the geometric properties of the transformed data, the study establishes criteria for the identification of mesoscale eddies based on these geometric attributes. The research utilizes sea surface flow field data sourced from the Copernicus Marine Environment Monitoring Service and validates the proposed algorithm through experimental comparisons with the traditional Velocity Geometry-based algorithm (VG algorithm). The findings indicate that the OG algorithm exhibits superior accuracy and computational precision in the detection of mesoscale eddies and in the calculation of each eddy's center when juxtaposed with the VG algorithm. Additionally, the OG algorithm not only excels in identifying standard eddies but also shows promising applicability in the detection of eccentric and dual-core eddies. Mesoscale eddies play a crucial role in ocean dynamics and significantly influence ocean circulation, heat transport, and ecosystems. Therefore, the development of a more efficient and precise mesoscale eddy detection algorithm holds substantial importance for advancing research in ocean dynamics and climate forecasting.

**Keywords:** orthogonal transformation; mesoscale eddy; eddy detection

## 1. Introduction

The mesoscale eddy is a fundamental component of oceanic dynamics, prevalent in national waters and significantly influencing the oceanic structure. Their study is crucial for deepening our understanding of marine environment dynamics, the structural and functional aspects of marine ecosystems, and the mechanisms driving oceanic climate change. Since the early 1970s, global oceanographers have been conducting research on mesoscale eddies, employing observational studies, numerical simulations, and theoretical analyses. In recent years, there has been a particularly vigorous focus on global ocean eddies, driven by advancements in remote sensing technologies and computational methodologies.

The system of mesoscale eddy detection methods has been continuously improved with the advancement of observational technology, and Dong et al. (2011) [1] classified

eddy detection methods into two paradigms, Eulerian and Lagrangian, based on data characteristics. Among them, Eulerian methods dominated the early studies, and Nencioli et al. (2010) [2] further divided them into three categories: methods based on physical parameters (e.g., vorticity, kinetic energy), detection methods based on geometrical features of the flow field, and a hybrid of the two. In the development of geometric characterization methods, the Winding-Angle (WA) method proposed by Sadarjoen and Post (2000) [3–7], which lays the theoretical foundation of geometric detection by calculating the streamline closure to detect eddy boundaries, is a milestone, and the Velocity Geometry-based (VG) algorithm is innovatively proposed by Nencioli et al. based on this method. Based on this approach, Nencioli et al. (2010) [2] innovatively proposed the Velocity Geometry-based (VG) algorithm, which improves the localization accuracy of eddy nuclei in complex flow fields by analyzing the spatial rotation characteristics of velocity geometry.

With the breakthrough of satellite remote sensing technology, the study of mesoscale eddies has entered a new phase of quantification, and the authoritative review by Chelton et al. (2011) [8] systematically integrates satellite observations and kinetic theories of global eddies, elucidating the central role of eddies in key processes such as energy cascading and mixing across isodensities, and providing physical constraints for algorithm design. Aiming to address the limitations of traditional geometric methods for detecting nonlinear eddy boundaries, Le Vu et al. (2018) [9] developed the Angular Momentum Eddy Detection Algorithm (AMEDA), which significantly improves eddy capture in a strong shear flow field by introducing an angular momentum parameter with streamline curvature analysis. Validation of the effectiveness of the algorithm benefits from the innovation of the data resources. The global daily-scale eddy dataset constructed by Faghmous et al. (2015) [10], which achieves standardized characterization of eddy trajectory, energy and other parameters through the fusion of multi-source satellite altimeters, provides a benchmarking platform for horizontal comparison of the algorithms.

At the level of dynamic tracking techniques, Mason et al. (2014) [11] broke the traditional static detection framework and proposed a tracking model based on the temporal and spatial evolution of the sea surface height field, which successfully resolved the nonlinear evolutionary behavior of eddy merging and splitting by constructing topological connectivity relationships during the eddy life cycle. These methodological advances have led to groundbreaking scientific findings: Zhang et al. (2014) [12] used multi-source observational data to quantitatively reveal that a single mesoscale eddy can transport up to 1/3 of the mass flux of the global meridional overturning circulation and empirically demonstrated the core mechanism of eddy matter transport from a kinetic perspective in the framework of Chelton theory.

At present, mesoscale eddy detection research has formed a synergistic development pattern of method innovation and data-driven mechanism analysis. The evolution from the winding angle method to the AMEDA reflects the paradigm shift in geometric detection from morphological description to dynamic feature extraction, while the combination of high-resolution datasets and tracking technology promotes the expansion of the research scale from static structural analysis to dynamic system modeling of eddy generation and dissipation processes.

This article introduces a novel orthogonal recognition algorithm predicated on geometric features. We conduct a comparative analysis of the Copernicus publicly available sea surface flow field data against the VG method to assess the efficacy of both algorithms, with the objective of optimizing the detection efficiency of various structured eddies and applying this methodology to eddy detection within the Indian Ocean. The flow field data utilized in this study is sourced from the Copernicus Marine Environment Monitoring Service (https://marine.copernicus.eu) (accessed on 1 July 2023), characterized by a spatial

resolution of $1/12 \times 1/12$ and $1/4 \times 1/4$ and a temporal resolution of once daily. The geographical scope of the study encompasses latitudes from 2S to 16N and longitudes from 80E to 98E, covering the period from 17 August 2024, to 18 August 2024. The other geographical scope of the study we have selected is from 25° N to 45° N, 60° W to 20° W; 0° S to 25° N, 55° E to 95° E; and 10° N to 40° N, 110° E to 150° E on 1 January, 1 April, 1 July, and 1 October.

## 2. Algorithm Comparison

### 2.1. Highlights

The article has the following two highlights:

Innovative algorithm design: A mesoscale eddy detection algorithm (OG algorithm) based on orthogonal geometric transformation is proposed to reconstruct the geometric features of flow field data through orthogonal transformation and significantly improve the recognition ability of complex eddy structures (such as dual-core and eccentric eddies).

Advantages of multi-type eddy detection: Experiments show that the OG algorithm has significantly higher detection rates (96.8%, 91.5%, 88.7%) for standard eddies, eccentric eddies and dual-core eddies than the traditional VG algorithm (92.4%, 78.2%, 46.8%), especially in asymmetric and small-scale eddy recognition.

By combining the geometric and dynamic characteristics of the flow field with the orthogonal transformation, this study breaks through the traditional method's dependence on the uniformity of the velocity field, solves the problems such as the detection of dual-core eddies, the missing detection of small-scale eddies and the positioning deviation of the boundary region, and provides a more accurate eddy detection tool for the study of marine dynamic processes.

### 2.2. VG Algorithm

You may be required to provide a graphical abstract at submission.

The detection of mesoscale eddies has been facilitated by recent technological advancements, leading to the development of a mesoscale eddy detection algorithm (VG algorithm) (2010) [2] that utilizes ocean surface current data. This algorithm delineates four constraints that align with the definition of the eddy velocity field and its associated characteristics, as shown in Figure 1. Points that meet all specified constraints are classified as each eddy center. The constraints are articulated as follows:

The east–west velocity component, denoted as, exhibits opposite signs on either side of each eddy center, with its magnitude increasing linearly as the distance from the center increases.

The north–south velocity component, represented as, similarly displays opposite signs on both sides of each eddy center, with its magnitude also increasing linearly with distance from the center.

The point of minimum space in the designated area is identified, which approximates each eddy center.

Near each approximated eddy center, the rotational direction of the velocity vectors must be consistent, indicating that the directions of two adjacent velocity vectors should either reside within the same quadrant or in two adjacent quadrants.

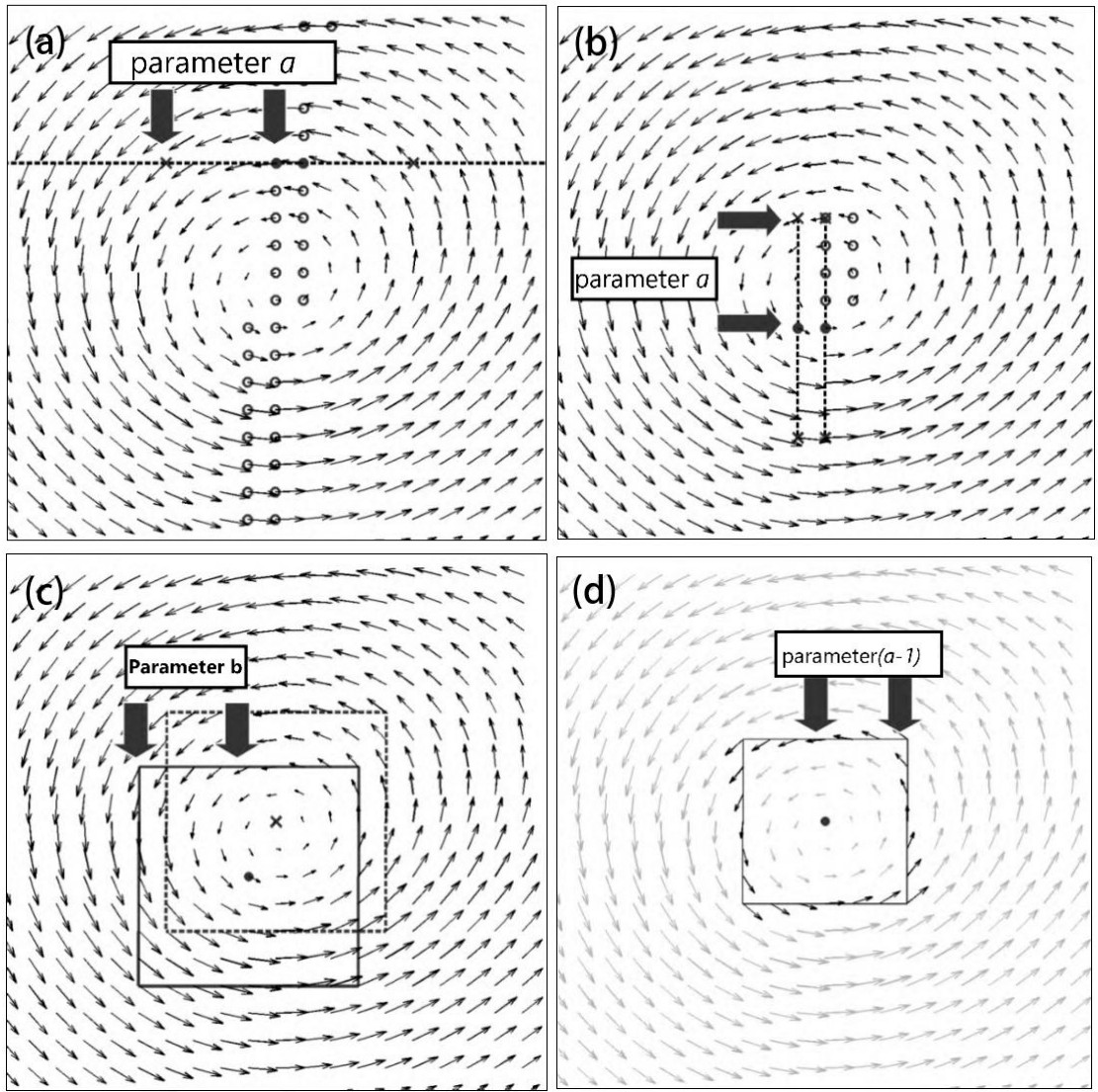

**Figure 1.** (**a**) Diagram of the first constraint of the VG algorithm, (**b**) Diagram of the second constraint of the VG algorithm, (**c**) Diagram of the third constraint of the VG algorithm, (**d**) Diagram of the fourth constraint of the VG algorithm. This schematic is adapted from [2] to illustrate the methodology applied in this study.

Within the established constraints, two parameters must be detected: one applicable to the first, second, and fourth constraints, and another specific to the third constraint. Specifically, parameter '*a*' as described in [2] is utilized to ascertain the number of grid points necessary for evaluating the increase in the east–west velocity component, denoted as $V_i$ (the first constraint). It similarly serves to determine the number of grid points required for assessing the increase in the north–south velocity component, $U_i$ (the second constraint). Furthermore, parameter '*a*' is instrumental in delineating the four boundary lines surrounding each eddy center, which are essential for analyzing changes in the direction of the velocity vector (the fourth constraint). Conversely, parameter '*b*' as described in [2] is designated for defining the extent of the local minimum velocity area (the third constraint). The algorithm allows for flexibility in the values of parameters '*a*' and '*b*', enabling their adjustment to accommodate the minimum scale of eddy detection, thereby enhancing the algorithm's applicability across grids of varying resolutions.

It is crucial to acknowledge that the magnitudes of parameters '*a*' and '*b*' correspond to the number of grid points, with the stipulation that '*a*' must be greater than or equal to 2,

while '*b*' must be greater than or equal to 1 and less than or equal to '*a*'. A reduction in the values of '*a*' and '*b*' enhances the algorithm's capacity to detect smaller-scale eddies. Although this algorithm demonstrates efficacy in pinpointing the locations of each mesoscale eddy center, the determination of detection thresholds remains a complex undertaking.

### 2.3. OG Algorithm

This article introduces an eddy detection algorithm grounded in geometric features, employing sea surface current data sourced from the Copernicus Marine Service for the detection of mesoscale eddies. The methodological steps are outlined as follows:

(1) Orthogonal Transformation: Initially, following the acquisition of the flow field data for analysis, an orthogonal transformation is applied to the north–south velocity component ($u_i$) and the east–west velocity component ($v_i$). To enhance the geometric signature of mesoscale eddies, we apply a fixed 90° counterclockwise orthogonal rotation to the original velocity field. The transformation is implemented using the standard 2D rotation matrix:

$$R(\theta) = \begin{bmatrix} cos\theta & -sin\theta \\ sin\theta & cos\theta \end{bmatrix} \tag{1}$$

where $\theta = 90°$. The original velocity components ($u_i, v_i$) are transformed into the rotated frame as:

$$\begin{bmatrix} U_i \\ V_i \end{bmatrix} = R(\theta) \cdot \begin{bmatrix} u_i \\ v_i \end{bmatrix} \tag{2}$$

Here, $U_i$ and $V_i$ denote the velocity components along and perpendicular to the principal axis, respectively.

Each vector undergoes an orthogonal transformation with $\theta = 90°$ according to Equation (1). This transformation converts the original vectors into a new north–south velocity component ($U_i$) and a new east–west velocity component ($V_i$), thereby forming a new ($U_i, V_i$) vector field. Initially, after acquiring the flow field data for analysis, an orthogonal coordinate transformation is applied to the original velocity components: the eastward component $u_i$ and the northward component $v_i$. Specifically, every velocity vector is rotated by a fixed angle of using the rotation matrix defined in Equation (1), yielding transformed components $U_i$ and $V_i$ as given by Equation (2). This produces a new velocity field ($U_i, V_i$) in which each vector has been rotated counterclockwise by 90° relative to its original orientation. Because a solid-body rotation (such as that of a mesoscale eddy) becomes purely radial after a 90° rotation, this transformation converts closed circulations into convergent or divergent patterns centered on the eddy core. As a result, potential eddy centers appear as local extrema (e.g., minima in speed or divergence) in the transformed field, while linear advection or shear-dominated flows—whose vectors do not form closed loops—do not produce such coherent focal structures. This effectively amplifies the signature of rotational features and suppresses non-rotational background flows. In practice, this step serves as the first stage of the Orthogonal Geometry (OG) algorithm: the raw velocity field (e.g., from satellite altimetry or numerical models) is uniformly rotated by 90° via Equations (1) and (2) as shown in Figure 2, and the resulting field is then scanned for geometric indicators of eddy presence, such as local velocity minima, directional convergence, and consistency in dot/cross product signs around candidate centers. This transformation enables clearer identification of eddy circulation patterns and structural symmetry.

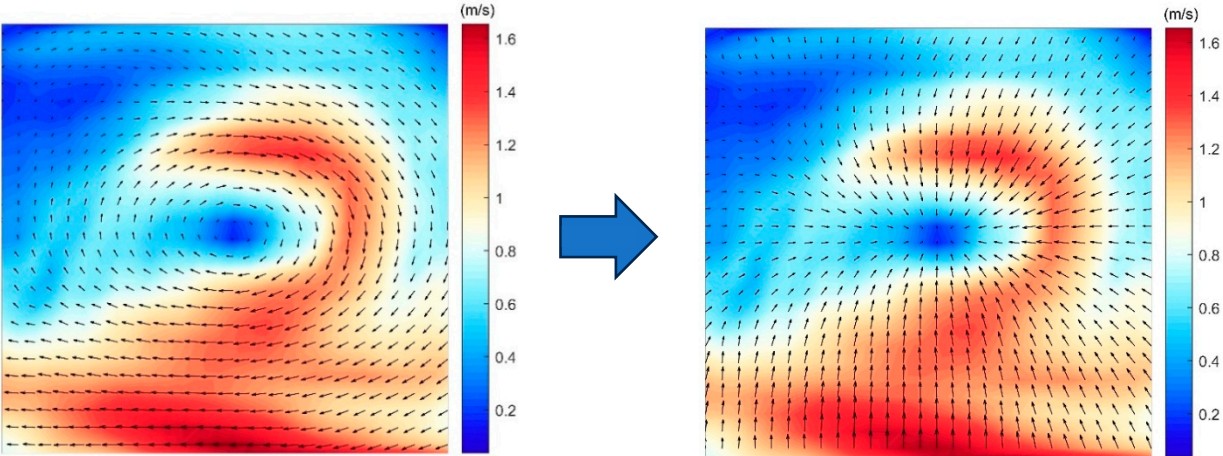

**Figure 2.** Effect of the 90° orthogonal transformation. Left: Original synthetic velocity field with a cyclonic mesoscale eddy (tangential flow). Right: Transformed field obtained by applying Equations (1) and (2) with $\theta = 90°$. The rotation converts tangential circulation into radial convergence/divergence, highlighting the eddy center.

(2) Preliminary screening of each eddy center: in consideration of the horizontal scale characteristics of ocean eddies, an appropriate window size ($11 \times 11$ in this study) is selected to systematically traverse the entire vector field (illustrated in (Figure 3.)). Utilizing the characteristic that the velocity at each eddy center is minimal, the point with the smallest absolute velocity value within each sliding window is designated as a potential each eddy center ($|U_i V_i|_{min}$). This step is designed to efficiently filter potential each eddy center locations.

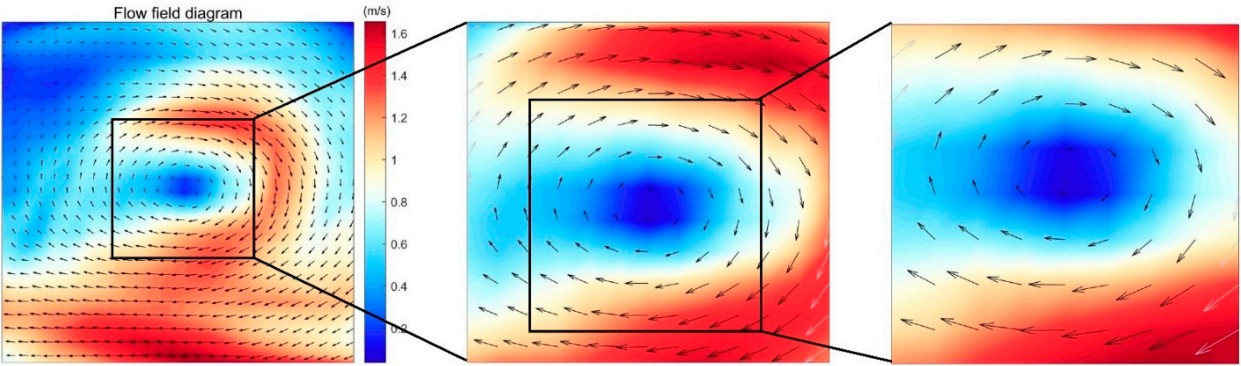

**Figure 3.** Schematic diagram of the flow field, where the black flow field is within an $11 \times 11$ grid. Schematic diagram of the flow field, where the black flow field is within a $7 \times 7$ grid.

(3) Refined positioning and verification: following the initial detection of each eddy center, a smaller sliding window is redefined using the golden ratio (with a size of $7 \times 7$ in this study), ensuring that the distance between the boundaries of the new window and the original window exceeds a predetermined distance threshold ($d = 3$, as depicted in (Figure 4). Additionally, the number of velocity direction vectors within the new window must surpass a preset threshold ($Q > 1$). To determine the precise location of each eddy center, we employ a least-squares fitting method based on the radial flow assumption. For each point $i$ ($1 < i < n$) in the $7 \times 7$ window, with coordinates $(X_i, Y_i)$ and transformed velocity components $(U_i, V_i)$, we assume that the velocity vectors radiate from or converge toward a common center $(C_X, C_Y)$. This leads to the linear relationship:

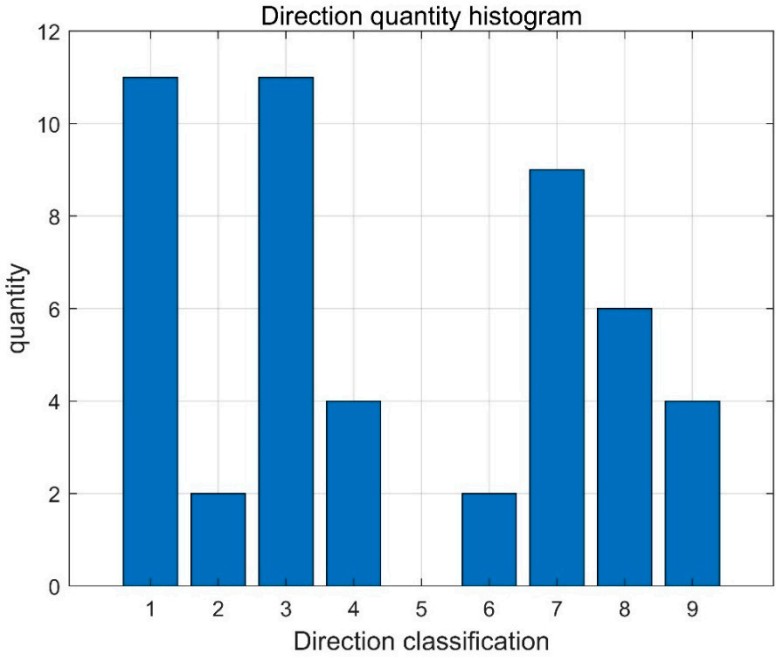

**Figure 4.** Bar chart representing the number of data points in the flow field for each direction.

$$C_X = X_i + t_i U_i, \ C_Y = Y_i + t_i V_i$$
$$C_X - t_i U_i = X_i, \ C_Y - t_i V_i = Y_i$$

where $t_i$ is an unknown scaling factor. Stacking these equations for all $i$ valid points lead to an over-determined linear system. Defining the unknown parameter vector as

$$X = \begin{bmatrix} C_X \\ C_Y \\ t_1 \\ \vdots \\ t_i \\ \vdots \\ t_n \end{bmatrix}$$

the system can be written in matrix form as:

$$AX = B \tag{3}$$

where the matrices $A$, $B$, and vector $X$ are structured as:

$$A = \begin{bmatrix} 1 & 0 & -U_1 & \dots & 0 & \dots & 0 \\ 0 & 1 & -V_1 & \dots & 0 & \dots & 0 \\ \vdots & \vdots & \vdots & \ddots & \vdots & \dots & \vdots \\ 1 & 0 & 0 & \dots & -U_i & \dots & 0 \\ 0 & 1 & 0 & \dots & -V_i & \dots & 0 \\ \vdots & \vdots & \vdots & \dots & \vdots & \ddots & \vdots \\ 1 & 0 & 0 & \dots & 0 & \dots & -U_n \\ 0 & 1 & 0 & \dots & 0 & \dots & -V_n \end{bmatrix}, \ B = \begin{bmatrix} X_1 \\ Y_1 \\ \vdots \\ X_i \\ Y_i \\ \vdots \\ X_n \\ Y_n \end{bmatrix}$$

In this formulation, each pair of consecutive rows in $A$, encodes the x- and y-coordinate constraints for a single observation point. The first two columns form an identity block

that isolates the unknown eddy center coordinates $(C_X, C_Y)$, while the remaining columns embed the negative velocity components $U_i$ and $V_i$ to model the linear coupling between position and flow direction. This structure enables the simultaneous estimation of the eddy center and the radial scaling factors $t_i$ through a unified linear framework. The optimal solution is obtained via the normal equations:

$$A^T A X = A^T B$$

yielding the closed-form estimate:

$$X = \left( A^T A \right)^{-1} A^T B$$

This approach leverages the inherent radial structure of coherent vortical motion to achieve sub-pixel eddy center localization without reliance on parametric flow models, thereby enhancing both the precision and robustness of the detection algorithm.

- To ensure the exclusion of each pseudo eddy center that is failed to satisfy the closure conditions, the following constraints are implemented:
- It is essential to verify that each flow field box contains at least one data point in each directional vector (as illustrated in (Figure 4));
- The grid points $A_1$, $A_2$, $A_3$, and $A_4$, located in the northeast, southeast, northwest, and southwest quadrants relative to each eddy center $(C_X, C_Y)$, are selected. The dot product of the vector extending from each eddy center to these four points with the directional vectors of these points is calculated by $\vec{A_i} \cdot \vec{X_i}$ by the gray and yellow arrows (as depicted in (Figure 5)). A notably small dot product value or one approaching $180°$ indicates that the vector direction of these points is nearly parallel to the vector connecting each eddy center to that point, suggesting an absence of conditions conducive to a closed flow field (as depicted in (Figure 5)).

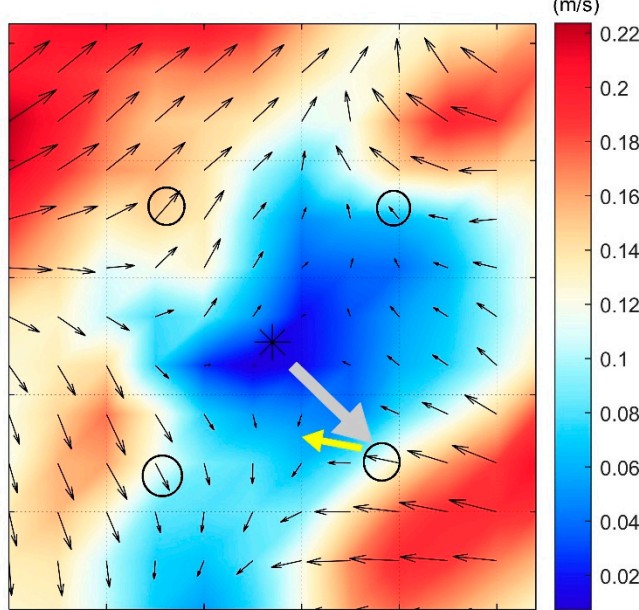

**Figure 5.** The figure shows the calculated flow field near each eddy center. The black crosses (*) indicate the calculated each eddy center, and the black circles (◯) denote the selected data grid points A1, A2, A3, and A4. The gray arrow and the yellow arrow calculate the inner product.

- The cross product $\vec{A_i} \times \vec{X_i}$ from each eddy center $(C_X, C_Y)$ to these four points is calculated as indicated by the gray and yellow arrows (as shown in (Figure 6)). Consistency

in the signs of the cross products signifies uniformity in the direction of the streamlines, thereby satisfying the closure condition. Conversely, if any point fails to meet this criterion, the corresponding each eddy center is excluded (as shown in (Figure 6)).

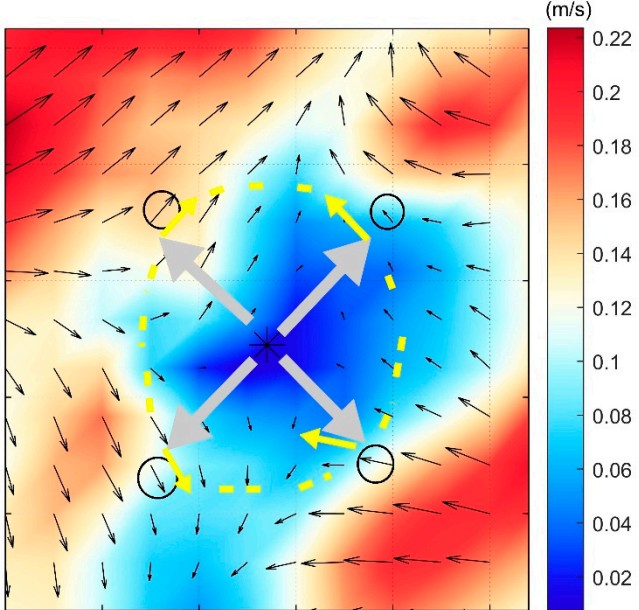

**Figure 6.** The figure shows the calculated flow field near each eddy center. The black crosses (*) indicate the calculated each eddy center, and the black circles (◯) denote the selected data grid points A1, A2, A3, and A4. Gray arrows and yellow arrows calculate the outer products.

## 3. Dataset-Based Evaluation

In order to assess the efficacy of the VG algorithm and OG algorithm in the detection of mesoscale eddies, we developed a comprehensive eddies dataset that encompasses both visual representations and NetCDF (.nc) files. This dataset allows for a systematic evaluation of eddy detection accuracy across diverse conditions.

### 3.1. Dataset Construction

The constructed dataset integrates multi-modal representations of eddy structures, including visual renderings (e.g., eddy morphology maps) and quantitative NetCDF files encoding velocity field data, enabling dual-core analysis of eddy dynamics. The graphical dataset offers a visual representation of eddy formations, while the NetCDF files preserve the fundamental velocity field data for subsequent quantitative analysis.

Eddies included in this dataset were generated and classified based on the following criteria:

Standard eddies: These are symmetric, circular eddies characterized by well-defined flow patterns.

Eccentric eddies: These eddies feature a core that is displaced from the geometric center, resulting in an asymmetric configuration.

Dual-core eddies: These complex eddies display a peanut-shaped or twin-core structure, often indicative of interactions between multiple eddies.

Each eddy sample is appropriately labeled, facilitating the evaluation of the performance of each algorithm across various eddy morphologies.

*3.2. Evaluation Methodology*

The dataset serves as a basis for evaluating the detection accuracy of two distinct methods across various eddy geometries. By implementing the VG algorithm and OG algorithm on this annotated dataset, we are able to quantify the following aspects:

The detection rate for each eddy category, defined as the proportion of accurately detected eddies.

The precision of each eddy center localization in instances where the eddies display atypical characteristics.

The capability to differentiate between single-core and dual-core eddies, which is a recognized limitation of conventional detection methodologies.

This organized dataset establishes a benchmark for assessing the efficacy of different eddy detection algorithms and their capacity to generalize across diverse eddy configurations. Different types of eddies pose distinct challenges for detection algorithms. The VG algorithm predominantly depends on constraints derived from the velocity field, while the OG algorithm leverages geometric properties and flow closure conditions, thereby facilitating a more adaptable detection process.

*3.3. Detection Accuracy Across Different Types of Eddies*

The efficacy of both algorithms was evaluated by examining the detection rate for each type of eddy. The detection rate is quantitatively defined as the proportion of accurately detected eddies relative to the total number of eddies present in the dataset. The findings are presented in Table 1.

**Table 1.** Detection Rates of VG Algorithm and OG Algorithm on Different Types of Eddies.

| Type of Eddies | Number of Sample 2 | VG Algorithm (%) | OG Algorithm (%) |
|---|---|---|---|
| Standard eddies | 200 | 92.4 | 96.8 |
| Eccentric eddies | 200 | 78.2 | 91.5 |
| Dual-core eddies | 200 | 46.8 | 88.7 |
| Overall | 600 | 72.5 | 92.3 |

The results suggest that the OG algorithm demonstrates superior detection accuracy for all categories of eddies, with particularly notable performance in the detection of eccentric and double-core eddies. Regarding standard eddies, the OG algorithm exhibits a marginal enhancement in accuracy, which may be attributed to the threshold selection of the VG algorithm, which appears to be inadequate for detecting smaller-scale eddies. Furthermore, in the context of asymmetric structures, the orthogonal transformation algorithm prioritizes the flow field's directional characteristics over rigid vector constraints, leading to improved performance outcomes.

*3.4. Each Eddy Center Localization Accuracy*

Given that each eddy center is delineated in geographical coordinates (latitude and longitude), we evaluate the precision of each algorithm by employing the Euclidean Distance Error (EDE). This metric quantifies the discrepancy between each detected eddy center and the established ground-truth center, expressed in kilometers, as shown in Table 2.

In summary, the eccentricity distance associated with the OG algorithm is less than that of the VG algorithm. When analyzing standard eddies, both methodologies exhibit minimal deviation in the detection of each eddy center, which suggests that the configuration of standard eddies is relatively symmetrical and stable, thereby facilitating accurate detection. The OG algorithm demonstrates an even smaller deviation distance, indicating

a higher precision in the recognition of standard eddies, whereas the VG algorithm may be susceptible to certain perturbations. In the case of irregular eddies, there is a notable increase in eccentricity distance, particularly for double eddy structures. This increase signifies that the intricate nature of dual-core eddy complicates the determination of a singular eddy center, resulting in greater errors. Although the deviation distance for the OG algorithm remains comparatively substantial, it is nonetheless more accurate than that of the VG algorithm, potentially due to its ability to partially account for the combined effects of the two cores.

**Table 2.** Each eddy center Localization Accuracy (Mean Euclidean Distance Error in km).

| Type of Eddies | VG Algorithm (km) | OG Algorithm (km) |
|---|---|---|
| Standard eddies | 1.2 | 0.8 |
| Eccentric eddies | 2.7 | 1.5 |
| Dual-core eddies | 7.4 | 4.2 |
| Overall | 3.7 | 2.2 |

*3.5. Spatial and Seasonal Generalization*

Figure 7 presents a comparative analysis of mesoscale eddy detection results between the Velocity Geometry-based (VG) and Orthogonal Geometry-based (OG) algorithms in the Northwest Pacific Ocean on four representative dates: 1 January, as shown in Figure 7(a1,a2), 1 April, as shown in Figure 7(b1,b2), 1 July, as shown in Figure 7(c1,c2), and 1 October 2024, as shown in Figure 7(d1,d2).

The VG algorithm relies on the closure and directional coherence of velocity fields, with detected eddies represented by red closed contours. This method performs well in regions with smooth flow fields and well-defined eddy structures, providing intuitive geometric boundaries and exhibiting low false positive rates. It is particularly suitable for identifying large-scale, regularly shaped eddies. However, in high-shear regions—such as the Kuroshio Extension—or under turbulent conditions, eddy boundaries often become fragmented or distorted, preventing the formation of closed streamlines. As a result, numerous small-scale or irregularly structured eddies are missed, leading to significant under-detection. This limitation is especially evident during high-energy seasons such as summer (July) and autumn (October), as shown in Figure 7c,d, where the number of detected eddies by VG is substantially lower than that of OG, indicating poor adaptability to complex flow regimes.

In contrast, the OG algorithm employs an orthogonal geometric transformation that enhances rotational features in the velocity field, enabling more effective identification of weak and localized vortices. Detected eddy centers are marked as red dots, with black asterisks indicating potential false positives. As shown in the Figure 7(a2–d2), the OG algorithm consistently identifies a denser and more spatially extensive distribution of eddies across all seasons. Notably, from spring (April) onward, it captures significantly more small-scale and fragmented eddies, particularly in dynamically active zones, demonstrating superior sensitivity and robustness in high-variability environments. Overall, the OG algorithm exhibits higher completeness, better temporal consistency, and greater adaptability to multiscale dynamics compared to the VG method.

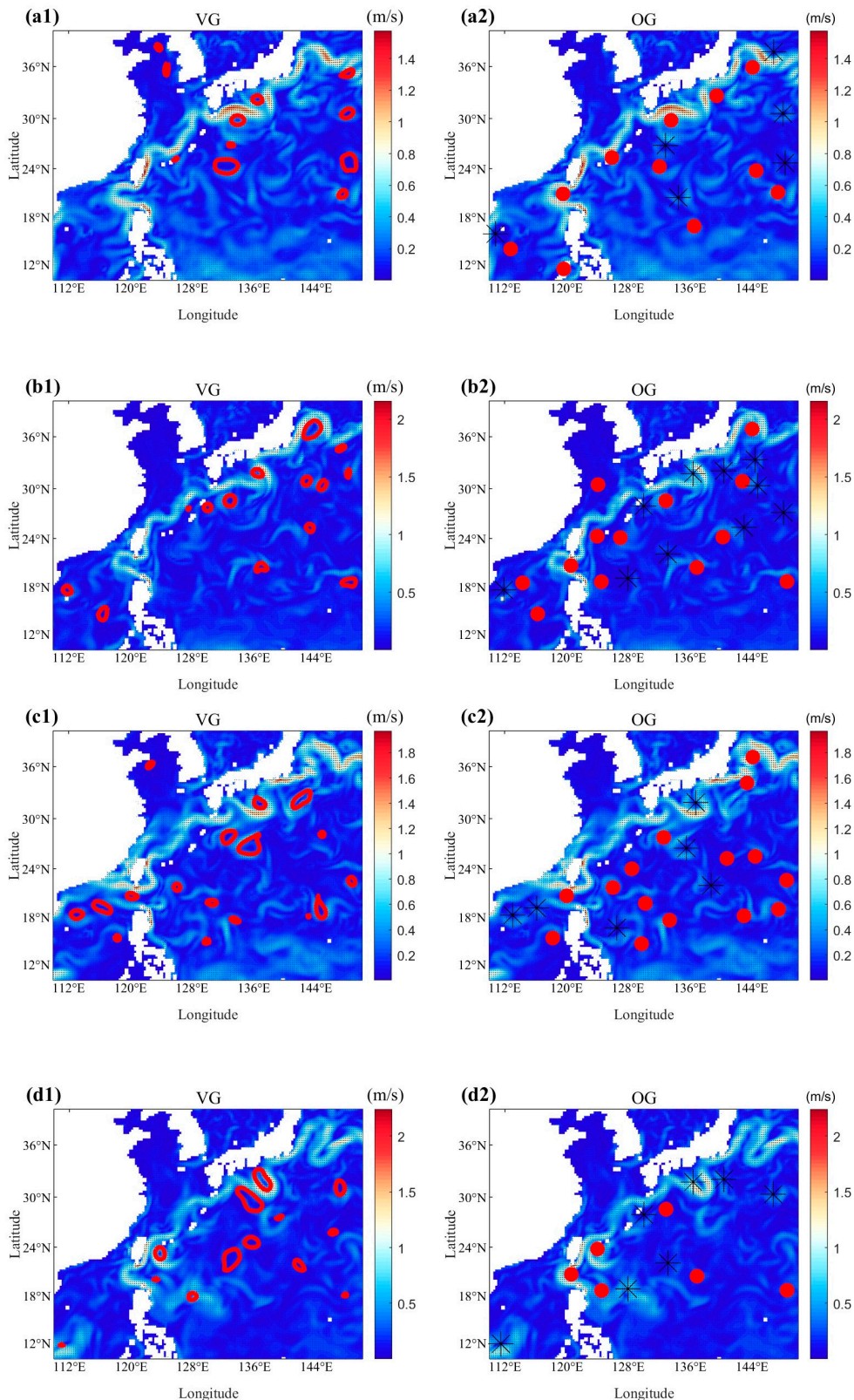

**Figure 7.** Comparison of mesoscale eddy detection results between the Velocity Geometry-based (VG) algorithm (**a1**–**d1**) and the Orthogonal Geometry-based (OG) algorithm (**a2**–**d2**) in the Northwest Pacific Ocean on four representative dates: 1 January, 1 April, 1 July, and 1 October 2024. The left detection results from the VG algorithm, with red contours indicating identified eddy boundaries. The right results from the OG algorithm, with red dots denoting detected cyclonic eddy centers and black cross (*) denoting anticyclonic eddy centers. Background color represents sea surface velocity magnitude (m/s).

Figure 8 compares mesoscale eddy detection results between the Velocity Geometry-based (VG) and Orthogonal Geometry-based (OG) algorithms in the North Indian Ocean across four representative seasons. The VG algorithm, relying on velocity field closure, identifies several well-organized eddies during winter and autumn but suffers from significant under-detection during spring and summer due to disrupted flow patterns under strong monsoon forcing. In contrast, the OG algorithm, as shown in Figure 8(a2–d2), enhances rotational features through orthogonal transformation and successfully detects a larger number of small-scale and coherent vortices throughout the year. The OG algorithm clearly resolves cyclonic–anticyclonic eddy pairs, particularly during the summer monsoon (July) in Figure 8(c2), indicating its ability to capture organized mesoscale structures driven by wind stress curl and shear instability. These results demonstrate that the OG algorithm not only improves detection sensitivity but also provides essential classification of eddy polarity, making it highly suitable for studying mesoscale dynamics in highly variable monsoon-influenced regions.

Figure 9 compares mesoscale eddy detection results between the Velocity Geometry-based (VG) and Orthogonal Geometry-based (OG) algorithms in the North Atlantic Ocean during four representative seasons: January, April, July, and October 2024. This region is dominated by the Gulf Stream Extension, one of the most turbulent and eddy-active zones in the global ocean. The VG algorithm, which relies on velocity field closure, successfully identifies several well-defined eddies during winter and autumn when flow patterns are relatively stable. However, during spring and summer—when intense turbulence and shear prevail—it frequently fails to detect many small-scale or deformed eddies due to disrupted streamlines, leading to substantial under-detection. In contrast, the OG algorithm (shown as Figure 9(a2–d2)) enhances rotational signals through orthogonal transformation and consistently detects a larger number of eddies across all seasons. These results demonstrate that the OG algorithm not only improves detection sensitivity but also provides essential information on eddy polarity, making it highly suitable for long-term eddy monitoring and climate-related studies in dynamically complex regions.

In summary, while the VG algorithm is suitable for qualitative analysis in stable flow conditions, the OG algorithm demonstrates superior performance in complex, high-energy oceanic environments. Its enhanced detection capability makes it more appropriate for large-scale statistical studies, eddy tracking, and oceanic process analysis, particularly in western boundary current regions characterized by intense mesoscale activity.

Table 3 presents a comparative analysis of the number of mesoscale eddies detected by the VG and OG algorithms across three oceanic regions during four representative seasons. Overall, the OG algorithm consistently identifies more eddies than the VG algorithm, with a total increase of 42.4%, indicating superior sensitivity and spatial coverage. The performance gap is most pronounced in the Northwest Pacific and North Atlantic Oceans, particularly during spring and summer, when strong shear and turbulent conditions lead to fragmented vortices that are difficult for the VG algorithm—relying on streamline closure—to detect. In contrast, the OG algorithm enhances rotational signals through orthogonal transformation, enabling robust detection even under complex flow regimes. Notably, in the North Indian Ocean during October, the OG algorithm detects fewer eddies than VG, possibly due to reduced monsoon forcing and a more stable flow field, suggesting region-specific sensitivity differences. These results demonstrate the superior robustness and adaptability of the OG algorithm across diverse oceanic environments, supporting its application in large-scale eddy monitoring and climate studies.

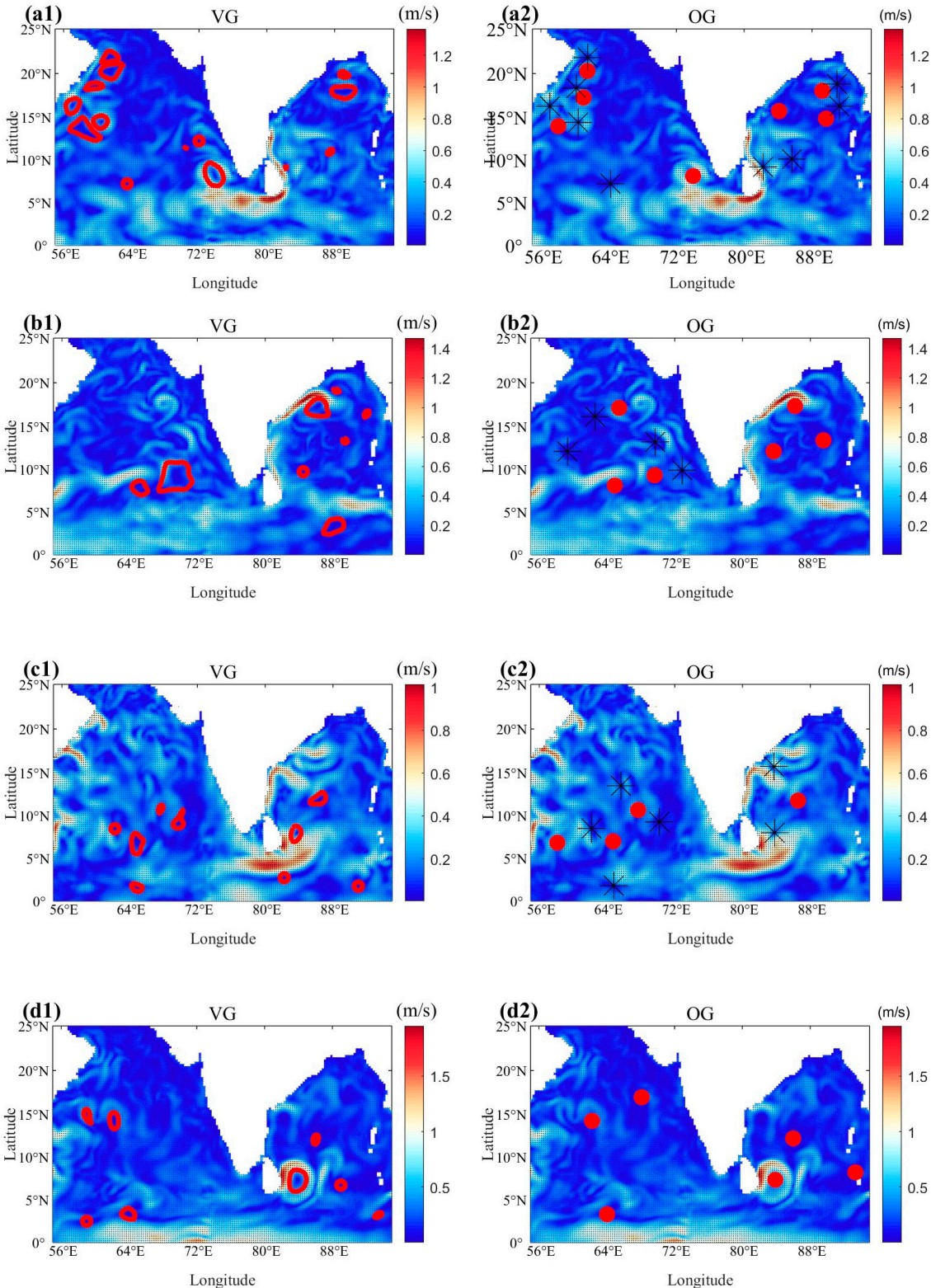

**Figure 8.** Comparison of mesoscale eddy detection results between the Velocity Geometry-based (VG) algorithm (**a1**–**d1**) and the Orthogonal Geometry-based (OG) algorithm (**a2**–**d2**) in the North Indian Ocean on four representative dates: 1 January, 1 April, 1 July, and 1 October 2024. The left detection results from the VG algorithm, with red contours indicating identified eddy boundaries. The right results from the OG algorithm, with red dots denoting detected cyclonic eddy centers and black cross (*) denoting anticyclonic eddy centers. Back-ground color represents sea surface velocity magnitude (m/s).

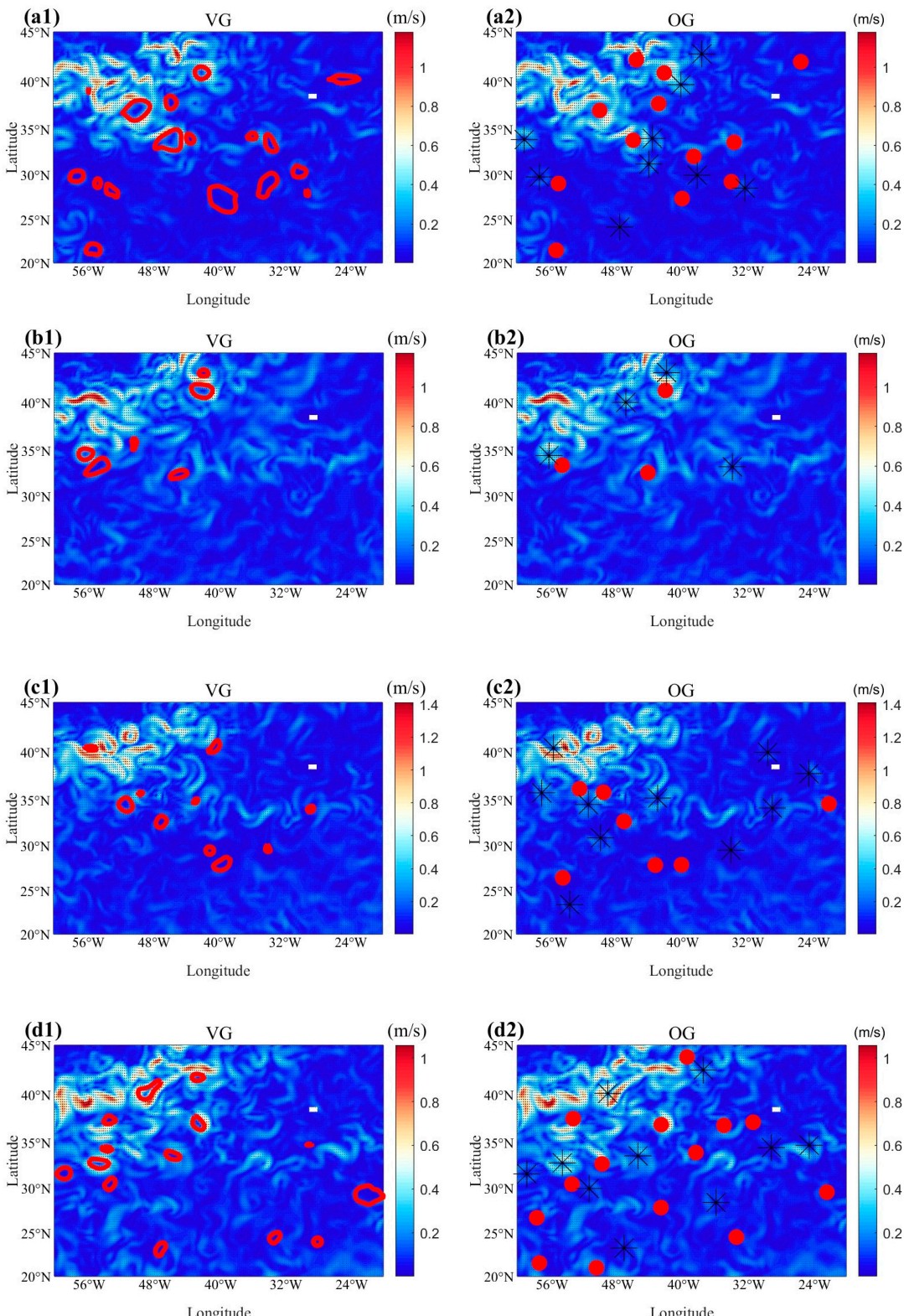

**Figure 9.** Comparison of mesoscale eddy detection results between the Velocity Geometry-based (VG) algorithm (**a1**–**d1**) and the Orthogonal Geometry-based (OG) algorithm (**a2**–**d2**) in the Atlantic Ocean on four representative dates: 1 January, 1 April, 1 July, and 1 October 2024. The left detection results from the VG algorithm, with red contours indicating identified eddy boundaries. The right results from the OG algorithm, with red dots denoting detected cyclonic eddy centers and black cross (*) denoting anticyclonic eddy centers. Back-ground color represents sea surface velocity magnitude (m/s).

**Table 3.** Number of Detected Mesoscale Eddies by VG and OG Algorithms in Three Oceanic Regions Across Four Seasons.

| Region | Date | VG Algorithm | OG Algorithm |
|---|---|---|---|
| Northwest Pacific | 1 January | 11 | 17 |
| | 1 April | 14 | 23 |
| | 1 July | 17 | 21 |
| | 1 October | 13 | 13 |
| North Indian Ocean | 1 January | 14 | 16 |
| | 1 April | 8 | 10 |
| | 1 July | 9 | 10 |
| | 1 October | 8 | 6 |
| North Atlantic Ocean | 1 January | 16 | 21 |
| | 1 April | 6 | 7 |
| | 1 July | 10 | 17 |
| | 1 October | 14 | 24 |

*3.6. Quantitative Sensitivity Analysis of Key Parameters*

To quantitatively evaluate the sensitivity of the VG and OG algorithms to their key configuration parameters, we select a representative case from the seasonal analysis in Section 3.5: the Northwest Pacific Ocean (10° N–40° N, 110° E–150° E) on 1 July 2023. This region exhibits intense mesoscale eddy activity associated with the Kuroshio Extension, providing a rigorous testbed for parameter robustness. The Sea Level Anomaly (SLA) used as a proxy for eddy cores is derived from the Copernicus Marine Service Global Ocean Ensemble Physics Reanalysis (product ID: GLOBAL_MULTIYEAR_PHY_ENS_001_031). Specifically, SLA is computed as the difference between the instantaneous sea surface height (zos-cglo) and its long-term temporal mean (zos-mean), both provided on a 0.25°. This ensemble-based reanalysis assimilates satellite altimetry, in situ temperature/salinity profiles, and sea ice observations, offering a dynamically consistent and observationally constrained representation of ocean variability. In the absence of manually labeled ground-truth eddy centers, we adopt local extrema in SLA as a widely accepted proxy for eddy core locations [8]. Anticyclonic eddies are matched to SLA maxima and cyclonic eddies to SLA minima within a 1.5° search radius. The Euclidean distance (in degrees) between each algorithm-derived center and its corresponding SLA extremum serves as the localization error metric.

For the VG algorithm, we test combinations of a∈{3,4,5} and b∈{2,3,4}. For the OG algorithm, we examine screening window sizes of 9 × 9, 11 × 11, and 15 × 15 coupled with refinement windows of 5 × 5, 7 × 7, and 9 × 9. The total number of detected vortices and their structural characteristics vary significantly across configurations. Specifically, the (a,b) = (3,2) setting yields 31 vortices, but many lack closed circulation contours, indicating susceptibility to noise or non-physical artifacts. In contrast, (a,b) = (5,4) detects only 6 vortices—structurally sound but likely missing moderate-intensity eddies. The intermediate pair (a,b) = (4,3) identifies 17 vortices, most of which exhibit well-defined, dynamically consistent closed loops.

Similarly, window size strongly influences detection performance. The smallest window pair (9 × 9/5 × 5) produces only 10 vortices, suggesting insufficient background separation. The largest pair (15 × 15/9 × 9) yields 18 vortices but with slightly blurred boundaries, while the 11 × 11/7 × 7 configuration detects 21 coherent vortices with optimal spatial definition and alignment with known mesoscale features in the region. Considering both vortex count and physical plausibility—particularly the presence of closed circulation structures and consistency with regional eddy dynamics—we select (a,b) = (4,3) for the

VG algorithm and window sizes of 11 × 11 (screening) and 7 × 7 (refinement) for the OG algorithm as the optimal configuration for subsequent analysis.

To further quantify the localization accuracy of each configuration, we compute the mean angular bias (in degrees) between algorithm-derived eddy centers and their matched SLA extrema. As shown in Table 4, the performance varies significantly across parameter settings. For the VG algorithm, the (a,b) = (3,2) configuration yields a relatively high mean bias of 0.472° despite detecting 10 vortices, indicating poor spatial precision likely due to sensitivity to noise. The (a,b) = (4,3) setting achieves a moderate bias of 0.458° with the same number of valid matches, suggesting improved robustness while maintaining detection capability. In contrast, (a,b) = (5,4) produces only 4 vortices but exhibits the lowest mean bias (0.319°), reflecting higher localization accuracy at the expense of reduced detection completeness.

**Table 4.** Mean angular bias (°) and number of valid matches between eddy centers detected by VG and OG configurations and SLA-derived reference centers.

| Method Category | Parameter Configuration | Valid Matches | Mean Bias (°) |
|---|---|---|---|
| OG Algorithm | 9 × 9/5 × 5 | 3 | 0.254 |
| | 11 × 11/7 × 7 | 13 | 0.292 |
| | 15 × 15/9 × 9 | 10 | 0.479 |
| VG Algorithm | a = 3 b = 2 | 10 | 0.472 |
| | a = 4 b = 3 | 10 | 0.458 |
| | a = 5 b = 4 | 4 | 0.319 |

For the OG algorithm, the smallest window pair (9 × 9/5 × 5) detects only 3 vortices but achieves the highest localization precision (mean bias: 0.254°), consistent with minimal smoothing effects. However, the low count suggests under-detection due to insufficient background separation. The largest window pair (15 × 15/9 × 9) increases the vortex count to 10 but results in a notably higher bias (0.479°), possibly due to over-smoothing that blurs eddy boundaries. The intermediate configuration (11 × 11/7 × 7) strikes an optimal balance: it detects 13 vortices—nearly twice as many as the smallest window—and maintains a competitive mean bias of 0.292°, demonstrating both high detection efficiency and accurate core localization.

These quantitative metrics reinforce our qualitative assessment: the (a,b) = (4,3) configuration for VG and the 11 × 11/7 × 7 window pair for OG not only yield physically plausible vortex structures but also exhibit superior alignment with SLA-derived reference centers, making them the most suitable choices for subsequent comparative analysis.

## 4. Results Analysis

### 4.1. Result Verification

On 17 August 2024, a comparative analysis was performed to evaluate the eddies detected by the VG algorithm in relation to those detected by the OG algorithm. The left image in Figure 10 depicts the eddies extracted by the VG algorithm on that date, while the right image illustrates the eddies detected through the OG algorithm. In these visual representations, black signifies cyclonic eddies, whereas red signifies anticyclonic eddies. The VG algorithm successfully identified a total of 12 eddies, in contrast to the 18 eddies detected by the OG algorithm. This comparison yields several insights: The OG algorithm recognized 7 additional eddies compared to the VG algorithm, yet it overlooked one eddy that was identified by the VG algorithm. Among the additional eddies detected by the OG algorithm approach, there was considerable variation in their morphology; specifically,

four were small-scale eddies spanning a few tens of kilometers, two were influenced by terrestrial or regional boundaries, and one had a dual-core structure resembling a peanut shape.

To facilitate a comprehensive evaluation of the performance of both algorithms across varying scenarios, four representative eddy cases were selected for detailed analysis. The Figure 10(a1–d1) illustrate eddy flow field images that were not successfully identified by the VG algorithm, while the Figure 10(a2–d2) display the locations of each eddy center detected by the OG algorithm under the same conditions.

In Figure 10(a1), the VG algorithm fails to detect mesoscale eddies due to its reliance on consistent velocity vector direction. The eddy exhibits a complex peanut-shaped structure, consisting of two independent internal circulations and a continuous, externally closed circulation. However, the VG algorithm struggles with such configurations because it requires a uniform directional pattern in velocity vectors. Near the velocity minima, the flow field is distributed across two distinct circulations, which prevents the algorithm from satisfying its detection constraints. As a result, the VG algorithm fails to capture the eddy's structure accurately, leading to incomplete or missed detection. In contrast, Figure 10(a2) illustrates the detection results obtained using the OG algorithm, which successfully detects the mesoscale eddy. By applying orthogonal transformation to the flow field, the OG algorithm effectively captures peripheral closed circulation and precisely detects each eddy center. Unlike the VG algorithm, which depends primarily on velocity vector alignment, the OG algorithm utilizes geometric transformations to assess flow field closure. This enables it to handle irregular and multicore eddy structures with higher accuracy. The ability of the OG algorithm to recognize dual-core and eccentric eddies further demonstrates its superiority over traditional methods.

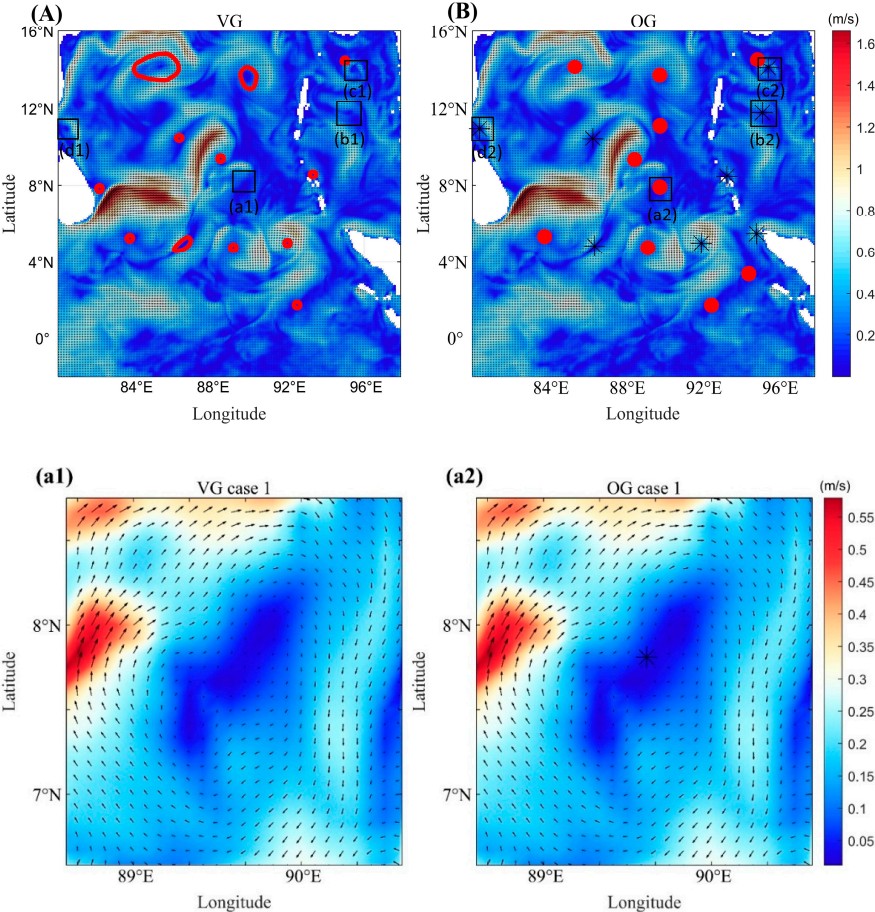

**Figure 10.** *Cont.*

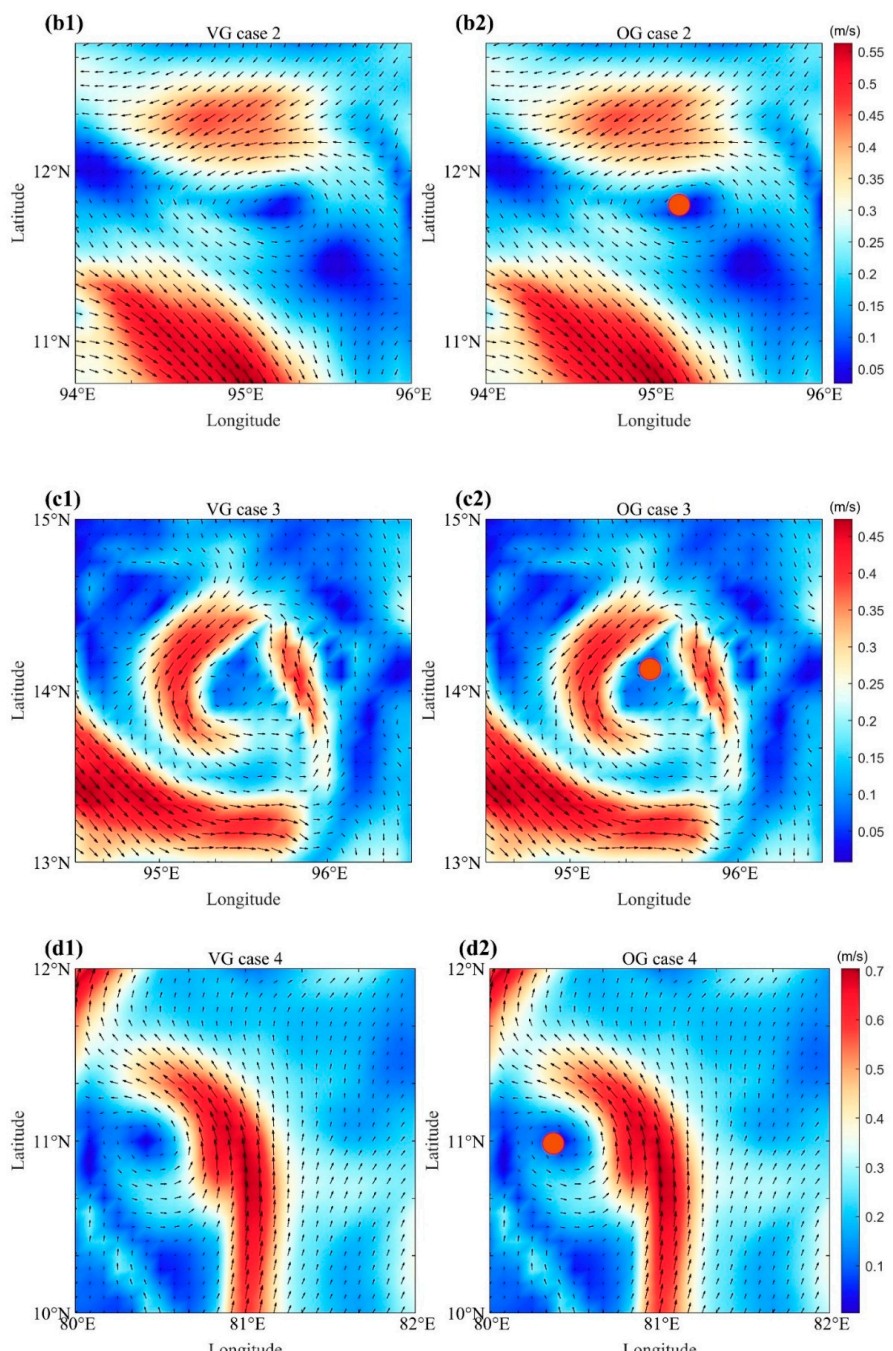

**Figure 10.** The base map is a flow field diagram. In area (**A**), the irregular red areas indicate the location and shape of the vortex identified by VG, area (**B**) represents the eddy center identified by OG. We have selected four representative areas (**a–d**), marked with red boxes. These selected areas are magnified and displayed in the subplots below. In these plots, red dots (●) are used to signify cyclonic eddies, and black cross (*) are used to signify anticyclonic eddies.

This comparison highlights the limitations of the VG algorithm in detecting complex eddy structures and underscores the advantages of the OG algorithm. By maintaining the geometric integrity of the flow field and ensuring consistent detection across various eddy structures, the OG algorithm provides a more accurate and reliable approach for mesoscale eddy detection, especially in cases where traditional methods face limitations.

In Figure 10b, the surrounding circulation of the eddies is robust; however, the eddy itself exhibits weak flow speed and a small spatial scale. The VG algorithm, which is based on the original sea surface geostrophic flow anomaly field and constrained by specific

parameter settings (a and b), encounters challenges in detecting such small-scale eddies. In contrast, the OG algorithm directly utilizes sea surface flow field data, encompassing all data grid points with a sliding window, which enables it to effectively capture the characteristics of small-scale eddies that are in the process of formation or dissipation. In Figure 10(b1), the VG algorithm fails to detect the mesoscale eddy due to its limitations in handling small-scale structures. Although the surrounding circulation remains robust, the eddy itself exhibits weak flow speed and a relatively small spatial scale. The VG algorithm, which is based on the original sea surface geostrophic flow anomaly field and constrained by specific parameter settings (a and b), struggles to detect such small-scale eddies. Since this algorithm relies on predefined thresholds and velocity gradients, it has difficulty detecting eddies in the early formation stages or those undergoing dissipation, where velocity gradients are less pronounced. Consequently, the VG algorithm either underestimates or completely fails to recognize the presence of small-scale eddies in this scenario. In contrast, Figure 10(b2) illustrates the results of the OG algorithm, which successfully detects the small-scale eddy. By directly utilizing sea surface flow field data and applying an orthogonal transformation, the OG algorithm analyzes the complete vector field without relying on strict velocity gradient constraints. Through a sliding window approach, it captures the local velocity minima and ensures that the detected eddy meets the geometric closure criteria. This enables the OG algorithm to effectively recognize small-scale eddies, even in cases where the flow speed is weak or the eddy structure is not well-defined. Additionally, the method proves advantageous in tracking eddies at different stages of evolution, including those in the process of formation or dissipation.

This comparison highlights the OG algorithm's superior capability in detecting small-scale eddies, overcoming the VG algorithm's limitations in detecting weak, low-gradient structures. By leveraging geometric transformations and comprehensive flow field analysis, the OG algorithm provides a more adaptive and reliable approach for detecting mesoscale eddies across varying spatial scales.

In Figure 10(c1), the eddy has an irregular shape, and while the central region shows partial closure, the flow field direction does not fully meet the VG algorithm's constraints, leading to missed detection. In contrast, Figure 10(c2) shows that the OG algorithm successfully detects the eddy by applying orthogonal transformation.

In Figure 10(d1), the VG algorithm fails to detect the eddy correctly, as it is located at the periphery of the study area where the flow field exhibits significant asymmetry. On one side of the eddy, the flow intensity is considerably higher than on the opposite side, resulting in an incomplete overall flow structure. The VG algorithm, which assumes a relatively uniform velocity gradient around each eddy center, struggles with such asymmetric conditions and fails to meet the criteria necessary for detection. This leads to missed or inaccurate detection of eddies near boundary regions. In contrast, Figure 10(d2) shows the detection results using the OG algorithm, which successfully captures the eddy despite its asymmetric structure. The OG method primarily emphasizes the directional characteristics of the flow field rather than strictly depending on uniform velocity gradients. In the initial determination of each eddy center, the algorithm considers exclusively the intensity and direction of the surrounding velocity vectors. This enables the recognition of eddies characterized by nonlinear intensity variations. Consequently, the OG algorithm effectively detects structurally incomplete eddies and those located in boundary regions, which are frequently overlooked by the VG algorithm.

This comparison underscores the limitations of the VG algorithm in detecting eddies near study area boundaries, where flow asymmetry is prominent. The original OG algorithm, by focusing on geometric transformation and directional coherence rather than strict

velocity constraints, provides a more robust and flexible approach for detecting eddies in complex flow environments.

Furthermore, there are instances where the VG algorithm erroneously classifies the eddy as possessing a closed flow field when it does not, as depicted in the red box in Figure 11. In Figure 11(a1), this occurrence suggests that the VG algorithm is highly sensitive to minor fluctuations in the vector field, which may result in the misclassification of data noise or non-eddy features as genuine eddies, particularly when the threshold is set too low. In contrast, in Figure 11(a2), the OG algorithm does not exhibit similar misclassifications, thereby demonstrating superior recognition accuracy and reliability.

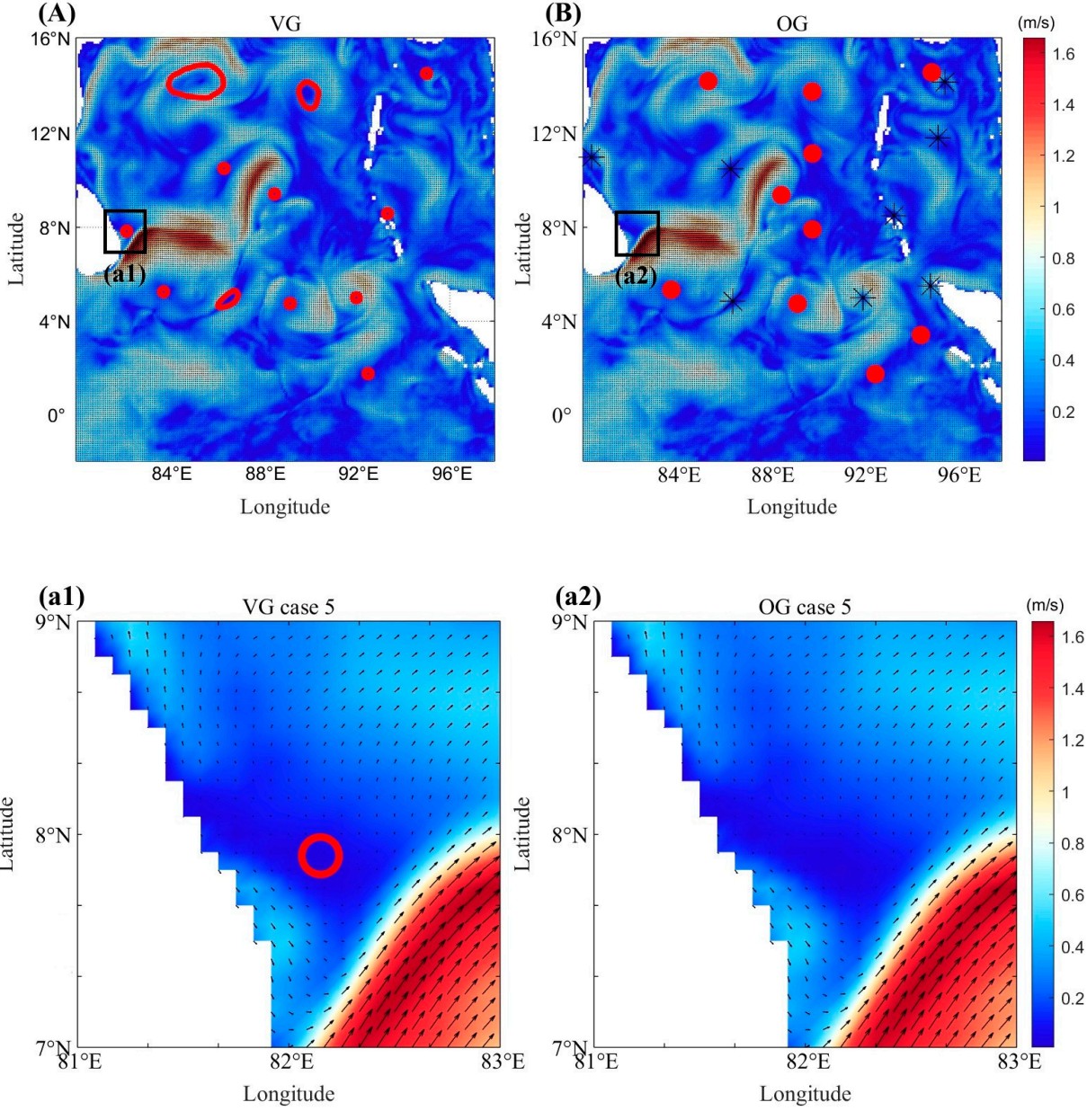

**Figure 11.** The base map is a flow field diagram. In the area (**A**) plot, irregular red regions indicate the location and shape of eddies. In the area (**B**), red dots (•) represent cyclonic eddies, while black dots (*) represent anticyclonic eddies. We have selected one representative area (**a**), marked with a red box. This selected area is magnified and displayed in the subplot on the right. In the magnified plot, red circles (O) are used to indicate eddies.

In conclusion, the orthogonal transformation method demonstrates enhanced capability and greater accuracy in detecting complex, small-scale, and boundary eddies when compared to the VG algorithm.

### 4.2. Comparison of Each Eddy Center

In the analysis of the detected eddies, the left figure (Figure 12) illustrates each eddy center as determined by the VG algorithm, while the right figure (Figure 12) presents each eddy center detected through the orthogonal transformation algorithm. A comparative examination reveals notable similarities between the two algorithms; however, discrepancies are also evident. Specifically, each eddy center detected by the VG algorithm is positioned lower and is situated closer to the region of the flow field characterized by higher velocity. In contrast, each eddy center detected by the orthogonal transformation algorithm is located precisely at the geometric center of each eddy. This variation may be attributed to the orthogonal transformation algorithm's process of reaffirming the eddies region surrounding the minimum point after initially detecting the minimum value of the eddies. In instances where the eddy size is diminutive, this algorithm is capable of excluding flow fields that do not pertain to the eddies in question. Consequently, the comparison of eddy center detection depicted in subsequent figures indicates that the orthogonal transformation algorithm has, to a certain extent, mitigated the influence of extraneous flow fields on the calculated position of each eddy center, thereby enhancing the accuracy of the determination of each eddy center.

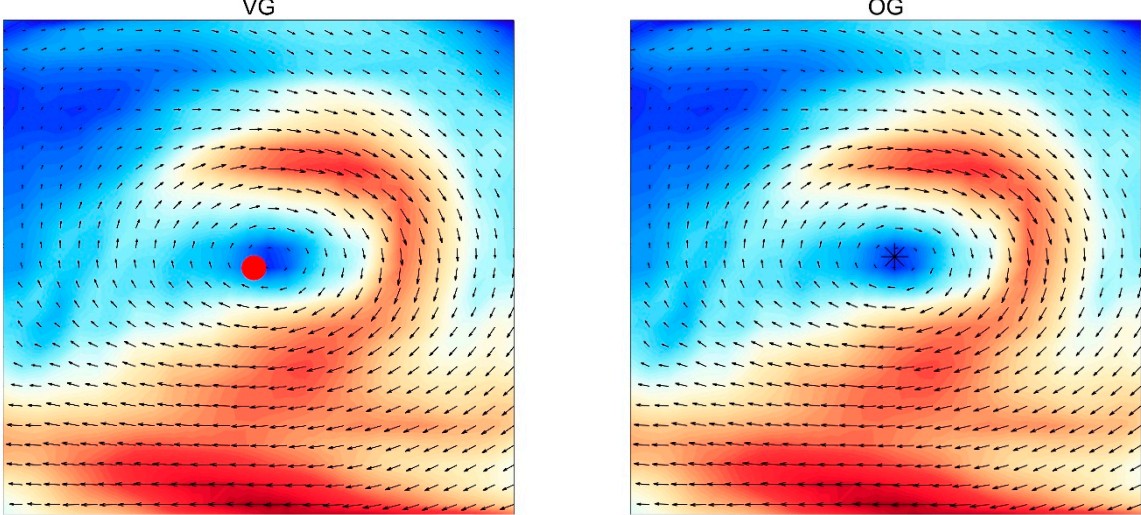

**Figure 12.** Each eddy center identified by the VG algorithm is represented by a red dot (●). Each eddy center detected by the orthogonal transformation algorithm is represented by a black cross (*).

### 4.3. Eddies Clustering

For irregular eddies, the VG algorithm still exhibits certain limitations. In contrast, the orthogonal recognition algorithm demonstrates superior performance in detecting such eddies. We applied this method to detect eddies with dual-core eddies and conducted cluster analysis on them. As illustrated in Figure 13, the detection results demonstrate that the top two eddies manifest a dual-core structure, while the bottom two exhibit a single-core configuration. Using the proposed method, we determined each eddy center. However, due to the connected and enclosed peripheral circulation of the top eddies, forming a larger, more regular flow pattern, they display a dual-core structure. Figure 14 presents the clustering results of the flow field in a three-dimensional space, where the X and Y axes represent spatial positions, the Z axis indicates velocity magnitude, purple dots

denote data points, and red dots mark cluster centers. As shown in the figure, the data points are distributed in two distinct regions, forming several well-defined clusters. The cluster centers, located at the heart of these clusters, represent each eddy center, indicating that they serve as representative points for each cluster. Figure 15 displays two different clustering outcomes. Each subplot is represented as a two-dimensional plane, with the *X*-axis and *Y*-axis denoting spatial positions. The initial cluster is predominantly situated in the lower portion of the image, thereby forming a high-density region. This observation serves to confirm a single-core eddy structure. In contrast, the second cluster is more dispersed in the upper part, containing two small dense regions, indicating a complex eddy composed of two smaller eddies, a dual-core eddy structure. Through the results of these two clusters, it is evident that the described flow characteristics differ: one represents a single large-scale eddy, while the other depicts a multi-scale complex flow pattern.

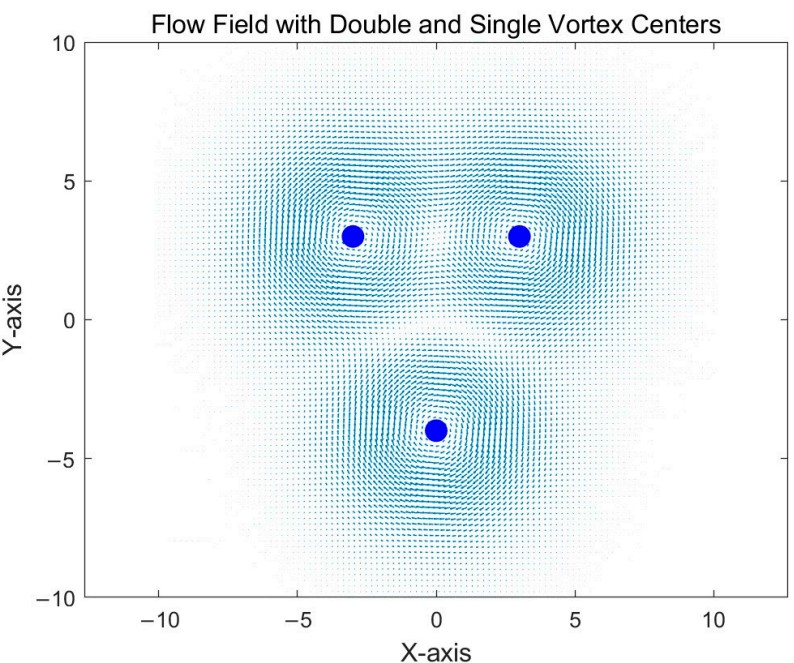

**Figure 13.** Diagram of an eddy flow field with dual-core eddy and single eddy.

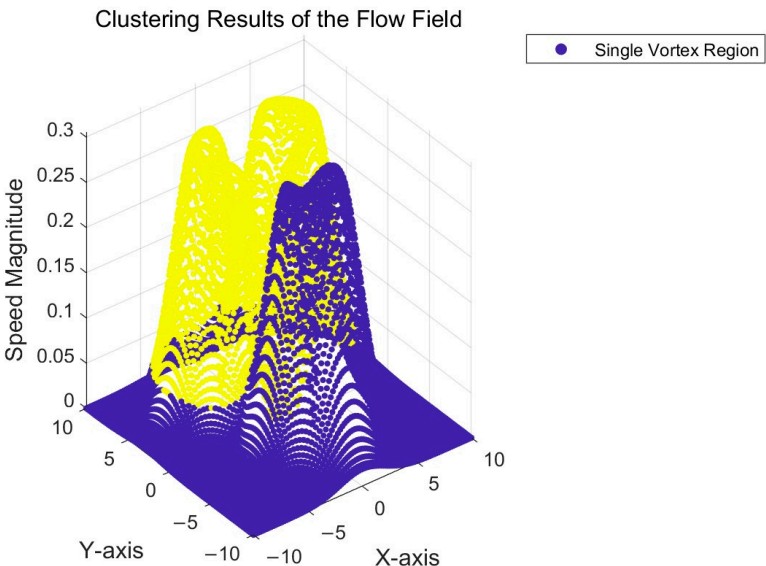

**Figure 14.** The result of clustering.

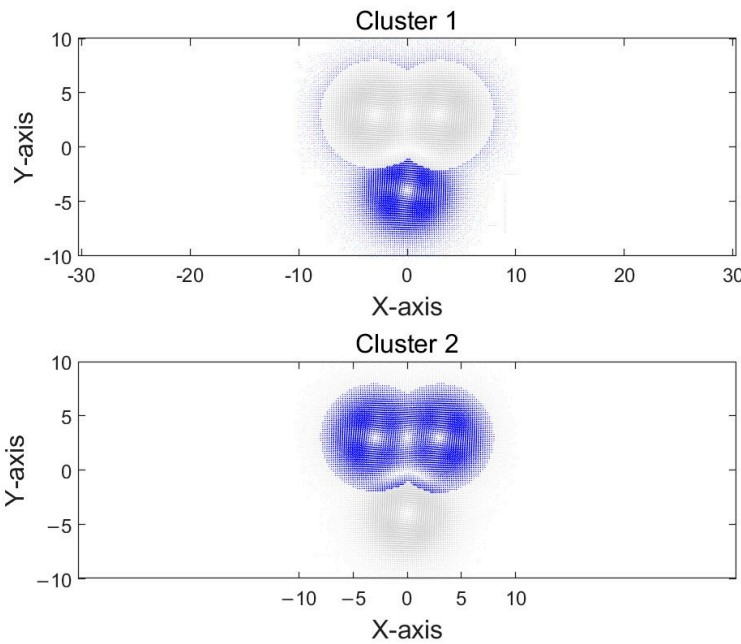

**Figure 15.** The result of clustering clusters.

## 5. Conclusions

This article proposes a mesoscale eddy detection algorithm based on orthogonal transformation of flow field data (OG algorithm). Based on the above comparative results, we draw the following conclusions:

Algorithm innovation: The OG algorithm combines orthogonal transformation with the least squares method to detect eddies from the dual perspectives of geometric structure and flow direction consistency.

The recognition effect is significantly improved: Compared to the VG algorithm, the OG algorithm not only performs better in the detection of standard eddies but also shows good applicability in the detection of eccentric eddies and dual-core eddies. Meanwhile, the OG algorithm exhibits smaller Euclidean distance errors in different types of eddies, improving the localization accuracy.

Adaptability to complex situations: The OG algorithm shows good applicability in eddy detection at small scales, incomplete boundaries, and structural asymmetry, addressing difficulties and errors encountered by the VG algorithm in detecting these eddies.

Overall, the OG algorithm successfully breaks through the bottleneck of the traditional method in asymmetric and non-standard eddy detection by introducing orthogonal transformation while retaining the advantages of the geometric detection principle and provides a new technological path for automatic detection and accurate extraction of mesoscale eddies, and also provides a more reliable tool for the research of ocean dynamics and climate modeling.

**Author Contributions:** Conceptualization, Y.C. and J.Y.; methodology, Y.C.; validation, Y.C., J.Y. and J.S.; formal analysis, J.Y.; resources, J.S.; data curation, J.Y.; writing—original draft preparation, Y.C. and J.Y.; writing—review and editing, Y.C. and J.Y.; visualization, J.Y.; supervision, J.S.; project administration, J.S.; funding acquisition, J.S. All authors have read and agreed to the published version of the manuscript.

**Funding:** This research and the APC was funded by Liaoning Province Science Data Center grant number [2025JH27/10100005]; Science and Technology Plan of Liaoning Province grant number [2024JH2/102400061]; Dalian Science and Technology Innovation Fund grant number [2024JJ11PT007]; Dalian Science and Technology Program for Innovation Talents of Dalian grant number [2022RJ06];

Liaoning Province Education Department Scientific research platform construction project grant number [LJ212410158039,LJ232410158056]. Basic scientific research funds of Dalian Ocean University (2024JBPTZ001,2024JBQNZ002).

**Data Availability Statement:** The new method database is not yet complete. If you need it, please feel free to contact us by email.

**Conflicts of Interest:** The authors declare no conflict of interest.

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
