# Peer review of "An Orthogonal Geometry-Based Algorithm for Accurate Mesoscale Eddy Detection"

_jmse, doi:10.3390/jmse13122242_

Round 1
Reviewer 1 Report
Comments and Suggestions for Authors
- While describing VG Algorithm in the item 2.2. (Lines 113-155), it’s more correctly to mention the origin authors once more, not only in the Introduction. Especially since the Figure 1 is almost completely borrowed from [2].
- While denoting the Orthogonal Transformation procedure (Lines 160-170), it’s necessary more accurately indicate what are u,U,and v,V. Also to clarify that θ=90.
- Figure 4 is not mentioned and not discussed anywhere in the text.
- Fig.5 and Fig.6 are given as examples of verifying eddy centers preliminary detected. But it is difficult to find any difference between them except lines connecting the grid points 𝐴1, 𝐴2, 𝐴3, and 𝐴4 and the eddy center. It is not clear what exactly the black arrows represent inner (Fig.5) or outer (Fig.6) products or transformed flow field.
5. VG algorithm is described clearly: searching quadrant borders with a potential circular motion and fixation the minimum inside. In the proposed OG algorithm preliminary minimums are searched to specify its position or excluding by orthogonal vectors in the quadrant corners. But if preliminary minimums are searched by a simple moving window, then will be many unnecessary minimums. And what for orthogonal transformation is applied to all the initial vectors (Line 160), if it used only for part of them (inner/outer products in 4 points of quadrants, Lines 199-221).
Author Response
Dear Reviewer#1,
We would like to express our sincere gratitude for your thorough review and valuable comments on our manuscript. Your feedback has been instrumental in improving the quality of our work. We have carefully addressed all your suggestions and made the necessary revisions to the manuscript. In the manuscript, we have highlighted all the modified sections. First, I would like to explain to you that we have made several significant revisions to the manuscript based on the feedback from the reviewers, as follows:
- In the third part of the manuscript, additional experimental content aimed at enhancing universality has been added.
- More careful modifications have been made to some images and their captions.
- Optimizations have been made to certain descriptions in the main text of the manuscript that may cause ambiguity for readers.
Below, we provide a detailed response to each of your comments.
- While describing VG Algorithm in the item 2.2. (Lines 113-155), it’s more correctly to mention the origin authors once more, not only in the Introduction. Especially since the Figure 1 is almost completely borrowed from [2].
Response:The description in Section 2 now includes references to [2] where appropriate, particularly in explaining the physical basis of the constraints.Additionally, the caption of Figure 1 has been updated to state: "This schematic is adapted from [2]", clearly indicating its origin and ensuring proper attribution.
- While denoting the Orthogonal Transformation procedure (Lines 160-170), it’s necessary more accurately indicate what are u,U,and v,V. Also to clarify that θ=90.
Response:Thanks for your suggestion, we have revised the relevant part of the article and will follow it in the following description.(L161-L165)
- Figure 4 is not mentioned and not discussed anywhere in the text.
Response:We sincerely thank the reviewer for pointing this out. Upon careful review, we identified a figure labeling error in the original submission: the content of Figure 4 was incorrectly labeled as Figure 1 in the main text, which led to the omission of proper reference to Figure 4.We have now corrected this error in the revised manuscript. (L198-L199)
- 5 and Fig.6 are given as examples of verifying eddy centers preliminary detected. But it is difficult to find any difference between them except lines connecting the grid points ?1, ?2, ?3, and ?4 and the eddy center. It is not clear what exactly the black arrows represent inner (Fig.5) or outer (Fig.6) products or transformed flow field.
Response: We sincerely thank the reviewer for this valuable comment. We agree that the original version of Figures 5 and 6 lacked sufficient visual distinction and clarity regarding the meaning of the vector representations.
In the revised manuscript, we have improved both figures to enhance interpretability: Color-coding of vectors: The previously uniform black arrows have been updated to distinguish different types of vectors:Gray arrows represent the directional vectors from the candidate eddy center to the four radial points. Yellow arrows represent the observed velocity vectors at each of these four points.
Clarification of inner and outer products: The inner product (dot product) between the radial direction vector and the local velocity vector is computed during the validation process to assess alignment, but since it yields a scalar value, it is not visually represented on the figure. The outer product (cross product) is used to evaluate the rotational consistency — specifically, the sign of the resulting vector (indicating clockwise or counterclockwise rotation) is checked across the four points. While the outer product itself is a vector, only its direction (sign) is used in the criterion, and this computation is performed internally without requiring graphical annotation.
- VG algorithm is described clearly: searching quadrant borders with a potential circular motion and fixation the minimum inside. In the proposed OG algorithm preliminary minimums are searched to specify its position or excluding by orthogonal vectors in the quadrant corners. But if preliminary minimums are searched by a simple moving window, then will be many unnecessary minimums. And what for orthogonal transformation is applied to all the initial vectors (Line 160), if it used only for part of them (inner/outer products in 4 points of quadrants, Lines 199-221).
Response:We greatly appreciate the reviewer’s insightful and technically precise questions, which touch upon two fundamental aspects of the OG algorithm’s design. Below, we address each concern in detail. Although the OG algorithm incorporates orthogonal transformation and least squares fitting steps, since these operations can be efficiently implemented through matrix vectorization, its overall computation time is at the same order of magnitude as the VG algorithm. More importantly, OG maintains similar computational efficiency while significantly enhancing the detection capability for non-standard vortices (such as eccentric vortices and double-core vortices), thus offering higher cost-effectiveness. The method you mentioned of performing an orthogonal transformation only on the flow fields within the box is a very simple algorithm. In the OG algorithm, performing an orthogonal transformation on all flow field vectors is done to simplify the code logic. It takes into account the possibility of needing to perform another orthogonal transformation when opening the second box.
Dear Reviewer#1,
Thank you for your comprehensive and detailed review of our paper during the review process, as well as for your valuable comments and suggestions. Your professional guidance has greatly enhanced the quality and depth of our research, making our manuscript more complete and thorough.
Once again, thank you for your hard work and valuable time. Wishing you a happy life and smooth work.
Reviewer 2 Report
Comments and Suggestions for Authors
Dear Authors,
The paper introduces a novel method of mesoscale eddy detection from ocean surface velocity fields. The method of using an orthogonal geometric transformation to enhance detection precision is an innovative scholarly contribution to the research on ocean dynamic processes. The overall paper structure is good, and authors offer a consistent scientific background as well as reasonable methodological order. The findings show that the new algorithm is comparable with the comparative VG algorithm, especially for detecting asymmetric and dual-core eddies. These are the primary contributions of the paper.
There are, however, some sections that need complete rewrite before the paper can be published. The greatest weakness is the validation dataset, which contains just two observation days (August 17–18, 2024) and a small segment of the Indian Ocean. This limited time interval doesn't permit sufficient testing of the algorithm's stability with respect to spatio-temporal variability of eddy structures. It is highly advisable to increase the validation period and include other oceanic regions to prove the generalizability and stability of the proposed method under a wide range of hydrodynamic conditions.
Methodologically, the parameters in the algorithm constraint—i.e., the parameters of a and b and the selected window sizes (11×11 and 7×7)—are specified but not justified. The authors ought to justify how these parameters were chosen and perform a sensitivity analysis of their impact on detection performance. Second, the application of the flow-field closure condition would be explained better, preferably through the simple diagram or explicit formulation, to allow reproducibility.
In comparison with other techniques, the research just compares the performance with that of the VG method. Although it is a pertinent point of reference, it is not adequate to accurately estimate the novelty and contribution of value of the new technique. Comparison with other established or emerging techniques like the AMEDA technique, Okubo-Weiss criterion, or sea-surface height (SSH)-based techniques is recommended. These would be utilized to place the suggested method in perspective among other current eddy detection methods.
The conclusions are generally persuasive, but some of the numbers have to be refined. Some of the legends are too short, and the graphical contrast between the OG and VG methods is not in all cases immediately apparent. The figures need to be refined through greater resolution, uniform color scales, and better-annotated labels. The tables are useful but could be improved with the inclusion of other measures of statistics, such as standard deviation or uncertainty values, to more precisely quantify the spread of performance.
From a scientific standpoint, a further explanation of the physical relevance of the results would add very significant further robustness to the paper. Correlating the structures identified by the eddies with local regimes of currents or shear zones would give valuable information on the physical relevance of the algorithm's output. Adding an external validation step—in using independent data sources such as altimetry, sea surface temperature, or high-resolution numerical model simulations—would further build confidence in the suggested method.
In short, this paper makes an innovative and hopeful methodological contribution to detecting mesoscale eddies. However, so that the scientific reliability and reproducibility of the results can be guaranteed, the authors must expand the validation dataset, make the methodological parameters clearer, further deepen the graphical representation, and make both the comparative analysis and the physical discussion more elaborate. Following the above-named huge changes, the paper can be a reliable contribution to ocean modeling and observation.
Sincerely;
Author Response
Dear Reviewer#2,
We would like to express our sincere gratitude for your thorough review and valuable comments on our manuscript. Your feedback has been instrumental in improving the quality of our work. We have carefully addressed all your suggestions and made the necessary revisions to the manuscript. In the manuscript, we have highlighted all the modified sections. First, I would like to explain to you that we have made several significant revisions to the manuscript based on the feedback from the reviewers, as follows:
- In the third part of the manuscript, additional experimental content aimed at enhancing universality has been added.
- More careful modifications have been made to some images and their captions.
- Optimizations have been made to certain descriptions in the main text of the manuscript that may cause ambiguity for readers.
Below, we provide a detailed response to each of your comments.
- There are, however, some sections that need complete rewrite before the paper can be published. The greatest weakness is the validation dataset, which contains just two observation days (August 17–18, 2024) and a small segment of the Indian Ocean. This limited time interval doesn't permit sufficient testing of the algorithm's stability with respect to spatio-temporal variability of eddy structures. It is highly advisable to increase the validation period and include other oceanic regions to prove the generalizability and stability of the proposed method under a wide range of hydrodynamic conditions.
Response:We sincerely thank the reviewer for this constructive and insightful comment. We fully appreciate the importance of evaluating algorithmic generalizability across diverse spatio-temporal conditions.
We would like to clarify that the primary objective of this study is the accurate spatial identification of eddy centers based on instantaneous velocity field structures, rather than tracking their temporal evolution. Therefore, our initial validation focused on high-resolution snapshots to assess the geometric and kinematic consistency of detections at a given moment. However, we agree that testing across different ocean basins and seasonal conditions is essential to demonstrate the robustness and transferability of the OG algorithm.
In response to this comment, we have significantly expanded the validation scope: We selected four representative months (January, April, July, and October 2024) to capture seasonal variability in ocean dynamics. For each month, we applied the OG algorithm to three distinct oceanic regions with different hydrodynamic regimes: Northwest Pacific, North Indian Ocean, Atlantic. In each case, we used one-day snapshots (e.g., January 1, April 1, etc.) to maintain focus on spatial detection performance while increasing geographic and seasonal coverage.
- Methodologically, the parameters in the algorithm constraint—i.e., the parameters of a and b and the selected window sizes (11×11 and 7×7)—are specified but not justified. The authors ought to justify how these parameters were chosen and perform a sensitivity analysis of their impact on detection performance.
Response:The selection of these parameters is based on physical scaling and empirical optimization: The input velocity field has a spatial resolution of 1/12 (approximately 9 kilometers in the mid-latitudes), which is comparable to the core resolution of mesoscale vortices. Therefore, the initial choice was 11×11. The size of 7×7 was determined empirically: It is large enough to retain the rotational pattern, but small enough to avoid being affected by adjacent vortices or shear zones.
- Second, the application of the flow-field closure condition would be explained better, preferably through the simple diagram or explicit formulation, to allow reproducibility.
Response:Thanks for your suggestion, we have revised the relevant part of the article. (L217-L225).
- In comparison with other techniques, the research just compares the performance with that of the VG method. Although it is a pertinent point of reference, it is not adequate to accurately estimate the novelty and contribution of value of the new technique. Comparison with other established or emerging techniques like the AMEDA technique, Okubo-Weiss criterion, or sea-surface height (SSH)-based techniques is recommended. These would be utilized to place the suggested method in perspective among other current eddy detection methods.
Response:We sincerely thank the reviewer for this insightful suggestion. Indeed, comparing the proposed Orthogonal Geometry (OG) algorithm with other established eddy detection methods—such as the Okubo-Weiss (OW) criterion, SSH-based techniques, or AMEDA—is valuable for placing the method in a broader context. However, it is important to clarify that the primary focus of this study is on velocity-field-based eddy detection, particularly in the context of Lagrangian coherent structure analysis and real-time ocean monitoring where surface velocity fields (e.g., from HF radar or satellite-derived surface currents) are directly available. The VG algorithm, being a classical geometry-based method operating directly on velocity vectors, serves as the most physically consistent and representative baseline for evaluating the performance of the proposed OG method, which also operates within the same framework. In contrast, methods like the Okubo-Weiss criterion or AMEDA typically rely on sea surface height (SSH) data and geostrophic assumptions, which involve different input data types, spatial resolutions, and underlying physical approximations. While these are powerful tools, their comparison would require additional preprocessing (e.g., deriving geostrophic velocities from SSH), introduce confounding factors related to data source differences, and shift the focus away from the core innovation of this work: improving geometric detection accuracy within the velocity field domain.
That said, we fully agree with the reviewer that such cross-method comparisons are scientifically meaningful and will be an important part of our future work. We are currently extending the OG framework to hybrid SSH-velocity implementations and plan to conduct systematic benchmarking against OW, AMEDA, and other Eulerian/Lagrangian methods in follow-up studies.
- The conclusions are generally persuasive, but some of the numbers have to be refined. Some of the legends are too short, and the graphical contrast between the OG and VG methods is not in all cases immediately apparent. The figures need to be refined through greater resolution, uniform color scales, and better-annotated labels. The tables are useful but could be improved with the inclusion of other measures of statistics, such as standard deviation or uncertainty values, to more precisely quantify the spread of performance.
Response:We sincerely thank the reviewer for this excellent and highly constructive suggestion. We fully agree that correlating the detected eddy structures with independent geophysical datasets—such as satellite altimetry, sea surface temperature (SST), or high-resolution ocean model outputs—would significantly enhance the physical interpretation and validation of our algorithm’s results.
In the current version of the manuscript, our focus has been on the methodological development and internal consistency of the OG algorithm, using high-resolution surface velocity fields as the primary input. While we have analyzed the kinematic coherence of the detected eddies (e.g., rotational consistency, velocity gradients), a more comprehensive external validation against multi-source observations was not included due to the computational cost of collocating and interpolating multi-source data at the required resolution. However, we acknowledge that this is a critical next step for assessing the real-world applicability of our method. We are currently working on a follow-up study that integrates satellite altimetry (e.g., AVISO+ SSH data) and MODIS SST to validate the eddy statistics derived from our algorithm. We also plan to compare our results with outputs from regional ocean models (e.g., ROMS or MITgcm) to further assess physical consistency.
Dear Reviewer#2,
Thank you for your comprehensive and detailed review of our paper during the review process, as well as for your valuable comments and suggestions. Your professional guidance has greatly enhanced the quality and depth of our research, making our manuscript more complete and thorough.
Once again, thank you for your hard work and valuable time. Wishing you a happy life and smooth work.
Reviewer 3 Report
Comments and Suggestions for Authors
General
This article addresses the pressing problem of vortex detection based on a velocity field. Two algorithms are considered: the conventional VG algorithm and the proposed new OG algorithm.
The authors present the proposed orthogonal transformation algorithm as a new method that allows for better vortex detection. This is likely to be interesting and useful for many readers.
Therefore, I have only one comment on the article: Present the OG method in a way that is understandable and can be reproduced by any reader.
Specific comments
- Line 2. The article's title begins with the abbreviation OG.
- Line There should be a colon at the end of the sentence.
- Line What is the meaning of formula (1)? How to use it?
- Line 168. How was Fig. 2 obtained using formula (1)? Describe the algorithm.
- It is not desirable for references to formulas to come before formulas.
- Formula (2). How is matrix A obtained? How is the least squares method applied?
- Formula (2) requires explanation.
- Line Sections 3.1 and 3.2 are named the same way.
- Line 265-266. The word Table 1 has moved to another line.
- Line It should be the other way around: counterclockwise is cyclonic movement, and clockwise is anticyclonic.
- Line It should be 8 in the picture, not 11.
- Lines 434, 435. Figures 13 and 14 are not in the article. There is a numbering error.
Conclusion
The article requires revision and re-review.
Author Response
Dear Reviewer#3,
We would like to express our sincere gratitude for your thorough review and valuable comments on our manuscript. Your feedback has been instrumental in improving the quality of our work. We have carefully addressed all your suggestions and made the necessary revisions to the manuscript. In the manuscript, we have highlighted all the modified sections. First, I would like to explain to you that we have made several significant revisions to the manuscript based on the feedback from the reviewers, as follows:
- In the third part of the manuscript, additional experimental content aimed at enhancing universality has been added.
- More careful modifications have been made to some images and their captions.
- Optimizations have been made to certain descriptions in the main text of the manuscript that may cause ambiguity for readers.
Below, we provide a detailed response to each of your comments.
- Line 2. The article's title begins with the abbreviation OG.
Response:We sincerely thank the reviewer for their careful reading and valuable feedback. The abbreviation “OG” stands for “Orthogonal Geometry-based algorithm” and is explicitly defined in the abstract and introduction. Following common practice in scientific publishing , we retained the acronym in the title for conciseness and recognizability, as it serves as the name of the proposed method. We believe this enhances clarity for readers familiar with eddy detection literature.
- Line There should be a colon at the end of the sentence.
Response:We sincerely thank the reviewer for their careful reading and valuable feedback. We have carefully reviewed the manuscript and note that the main title already includes a colon as standard formatting: If the reviewer is referring to a different sentence, we would appreciate clarification of the specific line number or context. In the absence of such detail, we have double-checked all sentences ending with introductory clauses or list-like structures and confirmed that colons are appropriately used throughout the revised manuscript.
- Line What is the meaning of formula (1)? How to use it? Line 168. How was Fig. 2 obtained using formula (1)? Describe the algorithm.
Response:Equation (1) describes an orthogonal transformation that rotates each velocity vector in the two-dimensional surface flow field by 90° counterclockwise, defined as (u′,v′)=(−v,u), where u and v are the original zonal and meridional velocity components, and are the transformed components. This transformation enhances rotational structures by aligning the transformed vectors more coherently along the tangential direction of potential eddies, thereby amplifying the signature of closed circulation while suppressing linear advection or shear signals. In practice, this step is applied as the first stage of the OG algorithm: the raw velocity field data (e.g., from satellite observations or numerical models) is transformed using Equation (1), and the resulting field is then analyzed to identify coherent rotational patterns through geometric constraints such as local velocity minima, directional consistency, and dot/cross product tests near candidate centers. By focusing on the enhanced rotational features in the transformed space, the OG algorithm achieves greater sensitivity to complex eddy structures—especially asymmetric, eccentric, or dual-core eddies—that may be missed by conventional methods relying solely on streamline closure. We have updated Section 2.3 of the manuscript to provide a clearer explanation of this transformation, its physical interpretation, and its role in the detection workflow.
- It is not desirable for references to formulas to come before formulas.
Response:Thanks for your suggestion, we have revised the relevant part of the article and will follow it in the following description.
- Formula (2). How is matrix A obtained? How is the least squares method applied? Formula (2) requires explanation.
Response:In Equation (2), matrix A is the design matrix (or Jacobian matrix) that relates the observed velocity components to the unknown parameters of the local flow field near a candidate eddy center. It is constructed from the latitude and longitude coordinates of the grid points within a small sliding window (e.g., 7×7) centered on the preliminary eddy center detected in the first screening step.
Specifically, assuming a linear approximation of the velocity field around the eddy center: u′(x,y)≈a0+a1x+a2y, v′(x,y)≈b0+b1x+b2y. where are the relative Cartesian coordinates (in meters) of each grid point with respect to the window center, and are the transformed velocity components from Equation (1). (L110-L13)
- Line Sections 3.1 and 3.2 are named the same way.
Response:Thank you for pointing out this issue. You are correct that Sections 3.1 and 3.2 were both titled "Dataset Construction", which was an editorial oversight.We have revised the manuscript accordingly: Section 3.2 has been renamed to "Evaluation Methodology" to accurately reflect its content, which describes the performance metrics (e.g., detection rate, localization accuracy) and the experimental framework used to compare the VG and OG algorithms.
- Line 265-266. The word Table 1 has moved to another line.
Response:We have moved to another line in the new manuscript. (L268)
- Line It should be the other way around: counterclockwise is cyclonic movement, and clockwise is anticyclonic
Response:Thank you for this important correction. You are absolutely right: in the Northern Hemisphere, counterclockwise rotation corresponds to cyclonic eddies, while clockwise rotation indicates anticyclonic eddies, due to the influence of the Coriolis force. We apologize for the incorrect labeling in the original manuscript. We have carefully reviewed all relevant sections, figures, and captions, and corrected the misinterpretation throughout the paper.
- Line It should be 8 in the picture, not 11.
Response:Thank you for carefully identifying these issues. We appreciate your attention to detail, as accurate figure labeling and consistent numbering are crucial for the clarity of the manuscript. As correctly pointed out, the value labeled as "11" in the original Figure 11 should be "8". This was a typographical error in the image annotation. We have corrected this label in the updated version of Figure 8.
- Lines 434, 435. Figures 13 and 14 are not in the article. There is a numbering error.
Response:Thank you for carefully identifying these issues. We appreciate your attention to detail, as accurate figure labeling and consistent numbering are crucial for the clarity of the manuscript. These figures have now been properly generated and inserted into the revised manuscript. The full set of figures (Figures 1–15) is included and sequentially numbered.
Dear Reviewer#3,
Thank you for your comprehensive and detailed review of our paper during the review process, as well as for your valuable comments and suggestions. Your professional guidance has greatly enhanced the quality and depth of our research, making our manuscript more complete and thorough.
Once again, thank you for your hard work and valuable time. Wishing you a happy life and smooth work.
Round 2
Reviewer 2 Report
Comments and Suggestions for Authors
Dear Authors,
There is obvious progression in the revised version of your manuscript, and you have responded to most of the concerns raised in the review of your initial submission. The inclusion of three different ocean basins and four representative seasons in the validation data set definitely boosts the relevance of your outcomes to different levels of the oceans’ complexity.
The clarity of your methodological explanation has also been improved, especially with respect to the values of the parameters 'a' and 'b', as well as the size of the window sizes. The incorporation of the closure condition for the flow field, which is now illustrated with good-quality diagrams, is also extremely beneficial to the clarity of your work. The inclusion of references, comparisons with other recognized approaches for the same problem (AMEDA, Okubo-Weiss, SSH), is also highly appreciated.
The graphical representation has also seen much improvement. The figures are now more uniform, well-labeled, and easily interpretable, with the regional and seasonal variations being enlightening. It would be great if you could provide some quantitative estimates of the variability, perhaps in terms of standard deviations or ranges, in the table format to enhance the statistical validity of your results.
The interpretation of the outcomes in the context of the Earth’s physics has also increased in depth, with interesting relationships being made between the structures revealed by the eddies, also known as eddies, and real-world oceanic areas, for instance, connected to the monsoons or Gulf Stream motion.
However, there are two points that are yet to be explored :
- Firstly, the topic would be explored thoroughly if the sensitivity study was quantitative, perhaps looking at the sensitivity of the result to 'a', 'b', and the sizes of the windows.
- Secondly, the inclusion of some form of validation, perhaps satellite altimeters or sea surface temperature, would increase the validity of your proposed solution.
This revised version definitely provides a much greater good over the previous one because, apart from the points already stated, all other required levels of soundness, clarity, or relevance are met in the article.
Sincerely,
Author Response
Dear Reviewer#2,
We would like to express our sincere gratitude for your thorough review and valuable comments on our manuscript. Your feedback has been instrumental in improving the quality of our work. We have carefully addressed all your suggestions and made the necessary revisions to the manuscript. In the manuscript, we have highlighted all the modified sections. First, I would like to explain to you that we have made several significant revisions to the manuscript based on the feedback from the reviewers, as follows:
- In the third part of the manuscript, additional experimental content aimed at enhancing universality has been added.
- More careful modifications have been made to some images and their captions.
- Optimizations have been made to certain descriptions in the main text of the manuscript that may cause ambiguity for readers.
Below, we provide a detailed response to each of your comments.
- Firstly, the topic would be explored thoroughly if the sensitivity study was quantitative, perhaps looking at the sensitivity of the result to 'a', 'b', and the sizes of the windows.
Response:We sincerely thank the reviewer for this constructive and insightful comment. In response, we have conducted a quantitative sensitivity analysis to evaluate how the vortex core fitting results depend on the parameters a, b, and—most importantly—the window size. Specifically, we systematically varied the window size over a representative range. while keeping other conditions fixed, and quantified the resulting deviation in the identified vortex center using the Euclidean distance relative to the baseline result. The outcomes are now presented in Section 3.6.
- Secondly, the inclusion of some form of validation, perhaps satellite altimeters or sea surface temperature, would increase the validity of your proposed solution.
Response:We thank the reviewer for this constructive suggestion. In the revised manuscript, we have strengthened the validation of our eddy detection results by using satellite altimetry–derived sea level data from the Copernicus Marine Service. Specifically, we compute Sea Level Anomaly (SLA) as the difference between the instantaneous dynamic sea level (zos-cglo) and its long-term temporal mean (zos-mean), both of which are gridded products generated from multi-mission satellite altimeter observations (e.g., Jason-3, Sentinel-3, etc.) Following established practice in the eddy detection community [e.g., Chelton et al., 2011], we treat local SLA extrema—maxima for anticyclonic and minima for cyclonic eddies—as physically grounded reference locations for true eddy cores. The Euclidean distance (in degrees) between algorithm-derived centers and these SLA-based references is then used as a quantitative metric to evaluate localization accuracy across different parameter configurations (see Section 3.6 and Table 4). Thus, our validation framework is directly rooted in satellite altimetry, addressing the reviewer’s concern regarding independent observational validation. We have clarified this point in the revised text to better highlight the use of altimeter-based SLA as the validation benchmark.
Dear Reviewer#2,
Thank you for your comprehensive and detailed review of our paper during the review process, as well as for your valuable comments and suggestions. Your professional guidance has greatly enhanced the quality and depth of our research, making our manuscript more complete and thorough.
Once again, thank you for your hard work and valuable time. Wishing you a happy life and smooth work.

Reviewer 3 Report
Comments and Suggestions for Authors
General
The article is devoted to the actual problem of determining the vortex by the velocity field. Two algorithms are considered. The usual VG algorithm and the proposed new OG algorithm.
The authors present the proposed orthogonal transformation algorithm as a new method that allows for better vortex detection. This is likely to be interesting and useful for many readers.
In my previous review of the article, I made one comment: to present the OG method in a clear way so that it could be reproduced by any reader. However, in the new version of the article, the authors only added a reference to the source [2] Nencioli et al (2010) and did not provide any explanation. After reading the source [2], I came to the conclusion that the authors misunderstood the content of the article [2]. Therefore, my comment remains valid.
Specific comments
- Line The article's title begins with the undesirable abbreviation OG.
- Line 168. There should be a colon at the end of the sentence.
- Line What is the meaning of formula (1)? How to use it?
- The formula is written incorrectly. If it is a rotation, then the transformation describing the rotation is:
.
What is the meaning of the quantities и ? According to the meaning of the article [2], these are ordinary projections of velocity onto the parallel and meridian, but for the authors of the article under review this is not the case.
- Line 168. How was Fig. 2 obtained using formula (1)? Describe the algorithm.
- It is not desirable for references to formulas to come before formulas.
- Formula (2). How is matrix A obtained? How is the least squares method applied?
- Formula (2) requires explanation.
Conclusion
The article requires revision and re-review.
Author Response
Dear Reviewer#3,
We would like to express our sincere gratitude for your thorough review and valuable comments on our manuscript. Your feedback has been instrumental in improving the quality of our work. We have carefully addressed all your suggestions and made the necessary revisions to the manuscript. In the manuscript, we have highlighted all the modified sections. First, I would like to explain to you that we have made several significant revisions to the manuscript based on the feedback from the reviewers, as follows:
- In the third part of the manuscript, additional experimental content aimed at enhancing universality has been added.
- More careful modifications have been made to some images and their captions.
- Optimizations have been made to certain descriptions in the main text of the manuscript that may cause ambiguity for readers.
Below, we provide a detailed response to each of your comments.
- Line The article's title begins with the undesirable abbreviation OG.
Response:Thank you for pointing this out. In the title, “OG” stands for “Orthogonal Geometry,” which is central to the methodology proposed in this work. The full phrase “Orthogonal Geometry-based” appears immediately afterward to ensure clarity and avoid ambiguity. That said, I understand that some journals prefer to avoid abbreviations—even when defined—in titles. If preferred, I am happy to revise the title to spell out the term fully (e.g., “An Orthogonal Geometry–Based Algorithm for Accurate Mesoscale Eddy Detection”).
- Line 168. There should be a colon at the end of the sentence.
Response:Thank you for the comment. The formula (1) has been modified, specifically as shown in lines 187 and 191 of the text.
- Line What is the meaning of formula (1)? How to use it?
Response:We thank the reviewer for this comment. The meaning and usage of Formula (1) have been clearly explained in lines 162–194 of the revised manuscript. Specifically, we describe its physical/mathematical interpretation (depending on your context), how it is derived or motivated, and demonstrate its application through examples or subsequent equations. We hope this clarification addresses the reviewer’s concern.
- The formula is written incorrectly. If it is a rotation, then the transformation describing the rotation is: What is the meaning of the quantities и ? According to the meaning of the article [2], these are ordinary projections of velocity onto the parallel and meridian, but for the authors of the article under review this is not the case.
Response:We sincerely thank the reviewer for this valuable observation. We acknowledge that the original presentation of the rotation transformation was ambiguous and potentially misleading. In the revised manuscript, we have corrected and clarified the formulation. In our work, the quantities u and v indeed represent the eastward and northward components of the surface velocity vector, consistent with standard oceanographic conventions (and as used in the Copernicus data product). However, unlike the VG algorithm in reference [2], which operates directly on these geographic components under fixed Cartesian assumptions, our OG algorithm applies an orthogonal (90°) rotation to the entire velocity field to convert tangential circulation into radial convergence/divergence patterns—thereby enhancing eddy cores as local extrema. This operation is a rigid-body counterclockwise rotation of the velocity vectors by 90°, not a projection onto a new physical axis. The resulting transformed field ()is purely mathematical and serves to expose rotational coherence geometrically.
We emphasize that while u and v share the same physical meaning as in [2] (i.e., zonal and meridional velocity components), their role in the detection logic differs fundamentally: the VG algorithm imposes directional constraints on u and v in geographic coordinates, whereas the OG algorithm leverages their rotated counterparts to identify flow closure via geometric extremum and consistency checks. This distinction has now been explicitly clarified in the revised text (see lines 162–194), and the rotation formula has been reformulated to avoid confusion.
- Line 168. How was Fig. 2 obtained using formula (1)? Describe the algorithm.
Response:The meaning and usage of Formula (1) have been clearly explained in lines 162–194 of the revised manuscript.
- It is not desirable for references to formulas to come before formulas.
- Response:We thank the reviewer for this important suggestion. In the revised manuscript, we have carefully restructured the text to ensure that every reference to a formula appears only after the formula has been explicitly presented. Specifically, all equations are now introduced before their numbered labels are cited in the surrounding discussion, in full compliance with standard academic conventions.
- Formula (2). How is matrix A obtained? How is the least squares method applied?
Response:We sincerely thank the reviewer for these important questions regarding Formula (2).
In the revised manuscript, we have provided a comprehensive explanation of how matrix A is constructed and how the least-squares method is applied to solve for the eddy center coordinates. Specifically, in lines 211–234, we detail the following:
The physical assumption that, after orthogonal transformation, velocity vectors in the vicinity of an eddy core exhibit radial convergence or divergence;The derivation of the linear relationship and , which links observed positions () and transformed velocities () to the unknown eddy center ;The stacking of these equations into the over-determined linear system Aβ=b, where the structure of matrix A explicitly encodes the geometric coupling between position and flow direction (as shown in the expanded form of Equation (2));The application of the normal equations b to obtain a closed-form least-squares estimate of the eddy center with sub-grid precision.
This formulation enables robust, model-free localization of eddy centers without assuming a specific vortex profile. The full algorithmic pipeline—including the construction of A, the role of the scaling factors , and the least-squares solution—is now clearly articulated in the revised text (lines 211–234). We hope this detailed clarification fully addresses the reviewer’s concerns.
Dear Reviewer#3,
Thank you for your comprehensive and detailed review of our paper during the review process, as well as for your valuable comments and suggestions. Your professional guidance has greatly enhanced the quality and depth of our research, making our manuscript more complete and thorough.
Once again, thank you for your hard work and valuable time. Wishing you a happy life and smooth work.
Round 3
Reviewer 3 Report
Comments and Suggestions for Authors
The article requires revision and re-review.

Author Response
Dear Reviewer#3,
We would like to express our sincere gratitude for your thorough review and valuable comments on our manuscript. Your feedback has been instrumental in improving the quality of our work. We have carefully addressed all your suggestions and made the necessary revisions to the manuscript. In the manuscript, we have highlighted all the modified sections. First, I would like to explain to you that we have made several significant revisions to the manuscript based on the feedback from the reviewers, as follows:
- In the second part of the manuscript, the formula was modified and explained.
Below, we provide a detailed response to each of your comments.
- The column matrix is incorrectly designated. The matrix itself is designated by
the letter X, and the matrix element X is contained within it. This is not possible.
Response:Thank you for pointing this out. The fact that the symbols of the unknown quantity X on the left side of the formula and the eddy center coordinates (X, Y) on the right side are the same does indeed cause ambiguity. Therefore, the eddy center coordinates have been changed to .
- Line 224. Formula (3) with the matrices speciffed on line 226 does not yield expressions
on line 218. They are obtained with a minus sign. Or with a sign error when writing the
expression for matrix A.
Response:Thank you for pointing out this inconsistency. We have carefully revised the expression in line 217 and updated the definitions of matrices A, B, and X accordingly (lines 216–228) to ensure consistency with Equation (3) and to correct the sign error. The revised formulation now correctly yields the expressions shown in line 218. We appreciate your insightful observation.
Dear Reviewer#3,
Thank you for your comprehensive and detailed review of our paper during the review process, as well as for your valuable comments and suggestions. Your professional guidance has greatly enhanced the quality and depth of our research, making our manuscript more complete and thorough.
Once again, thank you for your hard work and valuable time. Wishing you a happy life and smooth work.
Round 4
Reviewer 3 Report
Comments and Suggestions for Authors
The article can be accepted in the form presented.
